organic chemistry

cycloaddition, natural product, carbocycle

**Author for correspondence:**
Stefan Bräse
e-mail: braese@kit.edu

This article has been edited by the Royal Society of Chemistry, including the commissioning, peer review process and editorial aspects up to the point of acceptance.

# A versatile Diels–Alder approach to functionalized hydroanthraquinones

Janina Beck[1], Olaf Fuhr[2], Martin Nieger[4]
and Stefan Bräse[1,3]

[1]Institute of Organic Chemistry, Karlsruhe Institute of Technology (KIT), Fritz-Haber-Weg 6, 76131 Karlsruhe, Germany
[2]Institute of Nanotechnology (INT) and Karlsruhe Nano-Micro Facility (KNMF), and
[3]Institute of Biological and Chemical Systems-Functional Molecular Systems (IBCS-FMS), Karlsruhe Institute of Technology (KIT), Hermann-von-Helmholtz Platz 1, 76344 Eggenstein-Leopoldshafen, Germany
[4]Department of Chemistry, University of Helsinki, PO Box 55 (A.I. Virtasen aukio 1), 00014 Helsinki, Finland

The synthesis of highly substituted hydroanthraquinone derivatives with up to three stereogenic centres *via* a Diels–Alder reaction, starting from easily accessible 2-substituted naphthoquinones, is described. The [4+2]-cycloaddition is applicable for a broad range of substrates, runs under mild conditions and results in high yields. The highly regioselective outcome of the reactions is enabled by a benzoyl substituent at C2 of the dienophiles. The obtained hydroanthraquinones can be further modified and represent ideal substrates for follow-up intramolecular coupling reactions to create unique bicyclo[3.3.1] or -[3.2.2]nonane ring systems which are important natural product skeletons.

## 1. Introduction

Secondary metabolites produced by fungi, such as anthraquinone compounds, are known to possess a wide range of biological activities, including anti-cancer [1,2] antiviral [3,4] or antimicrobial activity [5]. Furthermore, anthraquinones exhibit chromatic properties enabling their use as dyes [6], are useful as chemical sensors [7] or organochelators [8]. Hydroanthraquinones as well are reported to inherit interesting properties such as cytotoxicity [9], antibacterial [10] and anti-cancer activity [11]. Nevertheless, only a few publications deal with synthetic approaches towards these attractive compounds [12–18].

Mycotoxins such as beticolins are natural products containing a hydroanthraquinone moiety linked to a chlorinated tetrahydroxanthone *via* a characteristic bicyclo[3.2.2]nonane ring system (figure 1) [19,20]. A diverse set of biological activities

R. Soc. Open Sci. **7**: 200626

**Figure 1.** The structure of *ortho*-beticolins [19,20].

such as antiproliferative effects on tumour cells and cytotoxicity owing to the formation of ion channels through cellular membranes are exhibited by these natural products [21–28]. The characteristic structure as well as the specific properties make these molecules intriguing to not only an organic chemist.

Given the manifold interesting properties of the described various anthraquinone compounds, efficient and reliable synthetic procedures towards diverse hydroanthraquinones need to be developed. Therefore, the use of the Diels–Alder (DA) reaction to build such attractive compounds from simple precursors is reported herein.

The well-known [4+2]-cycloaddition serves as a powerful and widely applied tool for introducing complexity in chemical structures; hence, it is important for the synthesis of natural products as well as new materials [29]. Quinones as dienophiles are the very first example investigated by Diels & Alder in 1928 [30]. These cyclic diones are synthetically useful and highly reactive molecules in pericyclic reactions and, therefore, one of the most important dienophiles for total synthesis applications [31]. Important carbocyclic scaffolds, more precisely hydroanthraquinones, are the typical products of DA cycloadditions with activated naphthoquinones and play an important role as building blocks in a large number of drugs and natural products [32,33].

The functionalized tetrahydroanthraquinone derivatives obtained in this study (table 1) represent valuable precursors for intramolecular couplings, such as palladium-catalysed Heck reactions, to create bicyclo[3.3.1] or -[3.2.2]nonane ring system scaffolds.

# 2. Results and discussion

## 2.1. Starting material synthesis

The studies were initiated with the straightforward synthesis of highly activated quinones by modifying a procedure developed by Buccini & Piggott [34]. The three-step synthesis provided 2-(2-iodobenzoyl)naphthalene-1,4-dione **5a** in excellent yield (scheme 1).

To synthesize similar 2-substituted naphthoquinones bearing functionalities, different benzoic acids **3a–f** were used in the acylation reaction (scheme 2, a). By introducing iodinated as well as brominated benzoic acids containing methoxy groups, hydroxy groups, *N*-acetyl or *N*-trifluoroacetyl residues, diverse quinone derivatives **5a–g**, which were supposed to function as dienophiles in following cycloadditions, were obtained in good yields of up to 84% over two steps.

The overall benefit of the two-step route from dimethoxynaphthalene **2** is its tolerance towards various functional groups and the scalability up to the multi-gram range.

X-ray analysis of naphthoquinone derivatives **5a–b** and **5d**, as well as their precursors **4a–f** confirmed the desired molecular structures with the naphthoquinone system being a planar structure. Selected examples are shown in figure 2 (see more in the electronic supplementary material).

## 2.2. Diels–Alder cycloadditions

By applying easily accessible precursors, the reaction conditions were investigated in a model reaction. For this purpose, solid 3-sulfolene **7** was suspended in high-boiling *o*-xylene and by heating the mixture to 125°C, gaseous 1,3-butadiene **6a** was released. The latter was led into a cooled reaction vessel containing either naphthoquinone **5a** or **5b** dissolved in dichloromethane. After warming the mixture to room temperature and subsequent stirring for 2 h, the iodinated **8aa** or the brominated anthraquinone derivative **8ba**, respectively, were isolated in good yields of up to 82% after flash chromatography on silica gel (scheme 3).

**Table 1.** Scope of the DA reactions between naphthoquinone derivatives **5a–g** and functionalized dienes **6a–h**. (Reaction conditions: argon atmosphere, dienophile (1.00 equiv.), diene (3.00–5.00 equiv.), CH$_2$Cl$_2$, 40°C, 3–5 h.)

**5a–g**    **6a–h**    conditions    **8aa–gh** *exo*    **9cd–gh** *endo*

| entry | dienophile | | diene | | product (yield (%)), {ratio of regioisomers} |
|---|---|---|---|---|---|
| 1 | X = I, R$^1$ = R$^2$ = H | **5a** | | **6b** | **8ab**[b,c] (63), {7.1 : 1} |
| 2 | X = Br, R$^1$ = R$^2$ = H | **5b** | | **6b** | **8bb**[b,c] (53), {7.7 : 1} |
| 3 | X = I, R$^1$ = R$^2$ = H | **5a** | | **6c** | **8ac**[a] (70) |
| 4 | X = Br, R$^1$ = R$^2$ = H | **5b** | | **6c** | **8bc**[a] (88) |
| 5 | X = I, R$^1$ = R$^2$ = H | **5a** | OTMS | **6d** | **8ad**[a] (20), **9ad**[a] (68) |
| 6 | X = Br, R$^1$ = R$^2$ = H | **5b** | OTMS | **6d** | **8bd**[a] (20), **9bd**[a] (61) |
| 7 | X = I, R$^1$ = R$^2$ = H | **5a** | OTMS | **6e** | **8ae**[a] (26), **9ae**[a] (48) |
| 8 | X = I, R$^1$ = R$^2$ = H | **5a** | OTBDMS | **6f** | **8af**[a] (29), **9af**[a] (38) |
| 9 | X = Br, R$^1$ = R$^2$ = H | **5b** | OTBDMS | **6f** | **8bf**[a] (27), **8/9bf**[d] (53) |
| 10 | X = I, R$^1$ = R$^2$ = H | **5a** | OTIPS | **6g** | **8ag**[a] (75) |
| 11 | X = I, R$^1$ = R$^2$ = H | **5a** | OTBDPS | **6h** | **8ah** (72) |
| 12 | X = Br, R$^1$ = R$^2$ = H | **5b** | OTBDPS | **6h** | **8bh**[a] (79) |
| 13 | Br, OMe | **5c** | | **6b** | **8cb**[b,c] (51), {10 : 1} |
| 14 | Br, OMe | **5c** | | **6c** | **8cc** (65) |

(*Continued.*)

| entry | dienophile | | diene | | product (yield (%)), {ratio of regioisomers} |
|---|---|---|---|---|---|
| 15 | Br / OMe | **5c** | OTMS | **6d** | **8/9cd**[d] (63), {5.6 : 1} |
| 16 | Br / OMe / OMe | **5d** | | **6b** | **8db**[b,c] (70), {6.7 : 1} |
| 17 | Br / OMe / OMe | **5d** | | **6c** | **8dc** (39) |
| 18 | Br / OH / OH | **5e** | | **6c** | **8ec** (54) |
| 19 | Br / N-H C(=O)CH₃ | **5f** | | **6b** | **8fb**[b,c] (38), {7.7 : 1} |
| 20 | Br / N-H C(=O)CH₃ | **5f** | | **6c** | **8fc**[a] (68) |
| 21 | Br / N-H C(=O)CF₃ | **5g** | | **6b** | **8gb**[b,c] (79), {7.7 : 1} |
| 22 | Br / N-H C(=O)CF₃ | **5g** | | **6c** | **8gc**[a] (84) |
| 23 | Br / N-H C(=O)CF₃ | **5g** | OTMS | **6d** | **8/9gd**[a,d] (76), {3.7 : 1} |

[a]Relative stereochemistry was determined by X-ray diffraction.
[b]The product was isolated as a non-separable mixture of diastereomers, (ratio of isomers as estimated by $^1$H NMR).
[c]For $R^4$ and $R^5$, two options are given since the exact structure could not be resolved by NMR spectra analysis.
[d]The exact structure was not resolved by NMR spectra analysis.

Analysis of the X-ray crystallographic data confirmed that the isolated products correspond to the novel tetrahydroanthraquinone scaffolds **8aa** and **8ba** bearing a halogenated benzoyl residue, in the solid state (figure 3). Two stereogenic centres have been created, including a sterically congested all-carbon quaternary stereocentre. The compounds represent interesting hydroanthraquinone structures which make suitable precursors for further functionalization reactions as well as intramolecular couplings.

Anthraquinone derivatives of higher complexity were obtained *via* DA cycloadditions between functionalized dienes **6b–6h** and highly activated 2-substituted naphthoquinones **5a–g**. All dienes **6a–h** subjected to cycloaddition reactions with the above-described naphthoquinone derivatives **5a–g**

**Scheme 1.** Synthesis of 2-(2-iodobenzoyl)naphthalene-1,4-dione **5a** from dihydroxynaphthalene **1**. Reagents and conditions: **a** MeI, $K_2CO_3$, DMF, rt, 20 h, 90%; **b** TFAA, reflux, 24 h, 67%; **c** CAN, $H_2O$/MeCN, −40°C/−20°C, 1 h, quant. DMF, *N,N*-dimethylformamide; TFAA, trifluoroacetic anhydride; CAN, ceric ammonium nitrate.

**Scheme 2.** Synthesis of naphthoquinones **5a–5g** and the corresponding yields over two steps. Reagents and conditions: **a** TFAA, reflux, 24 h, 65–84%; **b** CAN, $H_2O$/MeCN, −40°C/−20°C, 1 h, 12%–quant.

are shown in figure 4. The dienes contain different substitution patterns, including alkyl groups as well as silyl protected hydroxy groups with various steric demands.

Dienes **6b–d** were commercially available while trimethylsilyl-dienes **6e** and **6f** were synthesized from *trans*-2-methyl-2-butenal according to literature procedures [35,36]. Triisopropylsilyl (TIPS)-protected diene **6g** as well as tert-butyldiphenylsilyl (TBDPS) diene **6 h** were accessed *via* but-3-en-2-one [37].

In a typical DA reaction, the dienophile (1.00 equiv.) and the diene (3.00–5.00 equiv.) were dissolved in dry dichloromethane and the mixture was heated to 40°C. After completion of the reaction, as indicated by thin-layer chromatography (TLC) control, the solvent was removed under reduced pressure, and purification by flash chromatography on silica gel afforded the pure anthraquinone derivatives in yields of up to 88%.

The results of the reactions are summarized in table 1 with the general structures for the *exo* (**8**) and *endo* (**9**) products shown in the respective scheme. The regioselectivity of the DA reactions is described by the *ortho/meta/para* nomenclature with 1,2-disubstituted adducts named '*ortho*' as well as 1,4-adducts referred to as '*para*'.

(*a*) (*b*)

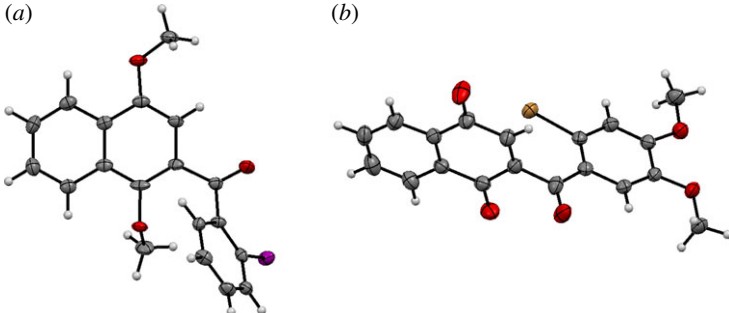

**Figure 2.** Molecular structures of the iodinated quinone **4a** (*a*), and the brominated dimethoxy dienophile **5d** (*b*). Displacement parameters are drawn at 50% probability level.

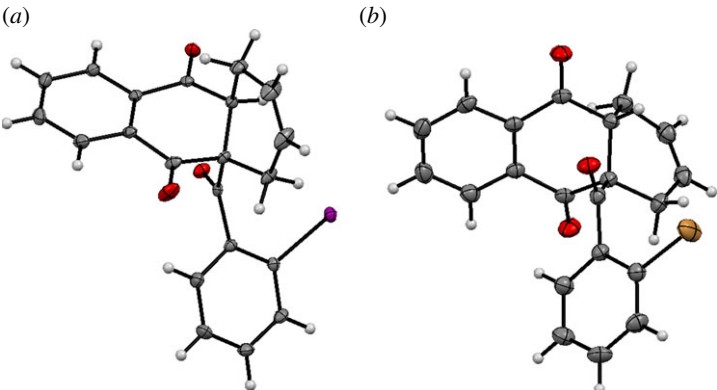

**5a**, X = I
**5b**, X = Br

**8aa**, X = I, 70%
**8ba**, X = Br, 82%

**Scheme 3.** DA model reaction with 3-sulfolene **7** and naphthoquinones **5a** and **5b** resulting in hydroanthraquinone derivatives **8aa** and **8ba**.

(*a*) (*b*)

**Figure 3.** Molecular structures of **8aa** (*a*) and **8ba** (*b*) determined by single-crystal X-ray diffraction. Displacement parameters are drawn at 50% probability level.

In general, the substrates underwent the cycloaddition very smoothly, under mild conditions and gave good to excellent yields. The cycloadditions with 1-substituted dienes resulted in diastereomeric mixtures; however, most of the reactions proceeded in a highly regioselective manner owing to the substituent at C2 of the dienophiles.

By using commercially available isoprene (**6b**), the DA reactions of both the iodinated **5a** and the brominated dienophile **5b** performed well and provided a mixture of regioisomers with one molecule occurring in large excess **8ab/8bb** (entries 1–2). The exact structure of the products could not be assigned with analysis of the nuclear magnetic resonance (NMR) spectra; however, the expected regioselectivity for the reaction with isoprene (**6b**) would give the 1,4-disubstituted '*para*' products **8ab** and **8bb** with $R^5$ being represented by the methyl group. Attempts to separate the regioisomers by flash chromatography on silica gel, preparative TLC on silica gel and preparative high-performance liquid chromatography were unsuccessful.

**Figure 4.** Dienes **6a–h** used for the DA reactions.

**Figure 5.** Molecular structure of **8bh** determined by single-crystal X-ray diffraction. Displacement parameters are drawn at 50% probability level.

To stepwise increase the complexity of the molecules, the reaction between 2,3-dimethyl-1,3-butadiene **6c** and dienophiles **5a** and **5b** was investigated next (entries 3–4). Here, excellent yields of up to 88% were obtained with the molecular structures of anthraquinones **8ac** and **8bc** being resolved by X-ray analysis.

For the incorporation of a protected hydroxy group into the anthraquinone core, TMS diene **6d** was used in the cycloadditions with **5a** and **5b** which gave the two diastereomers **8ad** (*exo*) and **9ad** (*endo*) (ratio 1 : 3.5) or **8bd** (*exo*) and **9bd** (*endo*) (ratio 1 : 3.0) which in each case were separated *via* flash chromatography on silica gel (entries 5–6). The reactions are proceeded by the usually high *endo* selectivity of DA reactions governed by the stereoelectronic nature of the reactants and owing to less steric clash in the *endo* transition state. The regioselectivity for the 1,2-disubstituted 'ortho' structure in all four obtained products (**8ad**, **9ad**, **8bd** and **9bd**) was confirmed by X-ray structure determination.

The application of methylated TMS diene **6e** in the DA reaction with iodinated dienophile **5a** provided the two 'ortho' diastereomers **8ae** and **9ae** (ratio 1 : 1.9) that could be separated *via* flash chromatography on silica gel (entry 7). Because the residue at position 1 of the dienes was the directing group in the reaction, exclusively the 'ortho' products were isolated, while no effect of the substituents at the diene on the regioselectivity of the reaction was observed. Single-crystal X-ray diffraction confirmed the selectivity of the reaction by resolving structures **8ae** and **9ae**.

By replacing the TMS group in the diene with a sterically more demanding tert-butyldimethylsilyl (TBDMS) group, as in **6f**, the complexity of the naphthoquinone products was further increased. The cycloaddition between dienophiles **5a** and **5b** and diene **6f** each afforded a mixture of two separable diastereomers (entries 8–9). The products were obtained with very similar yields and ratios of isomers in comparison to the reaction with trimethylsiloxy (OTMS) diene **6e** and, as the silyl ether acts as the directing group, the 'para' products were selectively formed. X-ray diffraction provided the molecular structure of the three products **8af**, **9af** and **8bf**. Empirical evidence suggests that the regioselectivity of the DA reactions with non-symmetrical dienes is predominantly governed by the electronic nature of the molecules, instead of steric effects.

The incorporation of a space demanding TIPS functionality in diene **6 g** resulted in a regioselective cycloaddition giving 'para' product **8ag** in 75% yield with its structure verified by X-ray crystallography (entry 10). Further regioselective cycloadditions owing to electronic reasons were observed in the reactions between TBDPS diene **6h** and dienophiles **5a** and **5b**. The iodinated anthraquinone **8ah** was isolated in 72% yield while the brominated product **8bh** yielded 79% with its structure verified by X-ray diffraction experiments (entries 11–12, figure 5).

To enable further modifications, such as the attachment of a tetrahydroxanthone moiety on the dienophiles, to facilitate anthraquinone–xanthone heterodimeric structures as found in beticolins (figure 1), it was envisioned to incorporate functionalities into the halogenated benzene ring of the

**Figure 6.** Molecular structure of **9gd** determined by single-crystal X-ray diffraction. Displacement parameters are drawn at 50% probability level.

naphthoquinone derivatives. Cycloaddition reactions between brominated dienophile **5c** bearing a methoxy group and isoprene (**6b**) as well as dimethylbutadiene **6c** gave a similar yield and ratio of products in comparison to the reactions with dienophiles **5a** and **5b** (entries 13–14).

The DA reaction between methoxy naphthoquinone **5c** and TMS diene **6e** resulted in a minor decrease in yield with the ratio of products shifted from 1 : 3.1 to 1 : 5.6 (entry 15), in comparison to the reaction of TMS diene **6e** with bromo dienophile **5b** (entry 6). This indicates an effect of the methoxy functionality on the selectivity of the reaction with one diastereomer forming in large excess. The exact structure of the products could not be verified by NMR spectra analysis. However, it is assumed that the same selectivity for the 'ortho' products with the major product being the endo tetrahydroanthraquinone is observed as in the reaction between **5b** and **6e**, as comparison of the NMR spectra gives the same characteristic signal pattern for the $CH_2$ group.

In addition, dimethoxy dienophile **5d** was employed in DA cycloadditions with isoprene (**6b**) as well as dimethyl diene **6c** (entries 16–17). By applying herein developed standard conditions, the reaction with **6b** resulted in an increased yield containing a non-separable mixture of products with a consistent ratio of regioisomers (6.7 : 1), in comparison to the cycloaddition between **5b** and **6b** (entry 2). In the reaction of dienophile **5d** with diene **6c**, only 39% of product **8dc** were obtained.

Cleavage of the methyl ethers of dimethoxy dienophile **5d** provided polar dihydroxy naphthoquinone **5e** which through a DA reaction with **6c** gave anthraquinone **8ec** in moderate yield (entry 18).

Via a cycloaddition reaction of dienophile **5f** bearing an N-acetyl residue with isoprene (**6b**), naphthoquinone **8fb** was obtained with 38% yield as a non-separable mixture of regioisomers in a ratio of 7.7 : 1 (entry 19). The DA cycloaddition of dimethyl diene **6c** with N-acetylated naphthoquinone **5f** proceeded smoothly to afford **8fc** in good yield (entry 20).

Dienophile **5 g** with an N-trifluoroacetyl residue was employed in cycloadditions with various dienes. First, the reaction with isoprene (**6b**) gave trifluoromethylated anthraquinone derivative **8gb** in significantly improved yield in comparison to the reaction of **5b** and **6b**; however, again, a non-separable mixture of regioisomers (ratio 7.7 : 1) was isolated (entry 21). Additionally, dienophile **5 g** underwent a reaction with dimethyl diene **6c** to give **8gc**, which was successfully crystallized and its molecular structure identified, in a very good yield of 84% (entry 22). The DA reaction with TMS diene **6d** resulted in a diastereomeric mixture of products, which showed the expected regiochemistry (entry 23). 'Ortho' anthraquinone derivatives **9gd** and **8gd** were isolated in 76% total yield in an endo/ exo ratio of 3.7 : 1. The structure of the endo product **9gd** was verified by X-ray crystallography (figure 6).

The substituents at the dienophiles, in general, did not render a significant impact on the outcome of the DA reactions. In some cases, however, decreased yields were observed, presumably owing to steric hindrance, whereas for the cycloadditions with isoprene (**6a**), mostly improved yields were obtained, in comparison to the reactions with non-functionalized dienophiles.

## 2.3. Modification of hydroanthraquinone **8ah**

When methods for the cleavage of the silyl ethers were examined, it was found that application of standard reagents such as tetrabutylammonium fluoride result in decomposition of the tetrahydroanthraquinones. By following a literature procedure for the cleavage of TBDMS ethers, it

**Scheme 4.** Attempted TBDPS ether cleavage of **8ah** towards **11** and the instead observed formation of acetal **10**.

**Scheme 5.** First intramolecular Heck reaction with **8aa** under standard conditions to give **12** whose molecular structure was verified by X-ray crystallography. Displacement parameters are drawn at 50% probability level. PMP, pentamethylpiperidine; DMA, dimethylacetamide.

was attempted to cleave TBDPS ether in **8ah** using a catalytic amount of acetyl chloride in dry methanol. Presumably, the reaction took place as expected; however, subsequent acid-catalysed addition of methanol occurred, resulting in modified hydroanthraquinone **10** bearing an acetal group (scheme 4).

## 2.4. Intramolecular coupling of anthraquinone **8aa**

To study the applicability of the synthesized anthraquinone derivatives for the construction of bicyclo[3.3.1] or -[3.2.2]nonane ring systems, an intramolecular Heck reaction under standard conditions was performed. By applying palladium acetate and triphenylphosphine, the reaction of **8aa** resulted in an anthraquinone bearing a novel [3.3.1]ring system **12** in good yield (scheme 5). Methods for the synthesis of [3.2.2]ring systems are currently under investigation.

# 3. Conclusion

*Via* a two-step route from dimethoxynaphthalene, the synthesis of various highly activated dienophiles applicable in [4+2]-cycloadditions was accomplished.

A DA approach facilitated straightforward access to highly functionalized anthraquinone derivatives by applying 2-substituted 1,4-naphthoquinones and various dienes in cycloadditions. Among these, a significant amount was analysed by single-crystal X-ray diffraction. The reactions tolerated a broad substrate scope, proceeded under mild conditions and resulted in good yields. The regiochemistry of the DA reactions was controlled by the benzoyl substituent at the dienophile, with 2-substituted dienes yielding 'para' hydroanthraquinones and dienes bearing substituents at C1 providing 'ortho' products. Moreover, the 'ortho' hydroanthraquinones were isolated as a diastereomeric mixture, favouring the sterically less hindered *endo* products, consistent with what was expected according to the *endo* rule. The results of this work suggest that the electronic rather than the steric nature of the substituents had the strongest influence on the regioselective outcome of the reaction. The incorporation of functionalities like methoxy groups into the anthraquinone derivatives paves the way for further modifications such as the installation of a tetrahydroxanthone subunit, for example, *via* a domino oxa-Michael–aldol condensation [38], to facilitate anthraquinone–xanthone heterodimers.

The hydroanthraquinone products of the DA cycloadditions comprise up to three stereogenic centres including a sterically congested all-carbon quaternary stereocentre and can be further modified, as

demonstrated exemplarily. The high potential of the obtained anthraquinones was demonstrated with the construction of a bicyclo[3.3.1]nonane ring system *via* an intramolecular Heck reaction.

# 4. Experimental procedure

## 4.1. General information

Reactions carried out under argon atmosphere were conducted using previously flame-dried glassware with standard Schlenk techniques. $^1$H NMR spectra were recorded on a Bruker Avance AV 300 (300 MHz) or a Bruker Avance 400 (400 MHz) as solutions at room temperature. Chemical shifts are expressed in parts per million (ppm, $\delta$) downfield from tetramethylsilane (TMS) and are referenced to CHCl$_3$ (7.26 ppm) as an internal standard. All coupling constants are absolute values and $J$ values are expressed in Hertz (Hz). $^{13}$C NMR spectra were recorded on a Bruker DRX 500 (126 MHz) spectrometer. Chemical shifts are expressed in parts per million (ppm, $\delta$) downfield from TMS and are referenced to CDCl$_3$ (77.2 ppm) as an internal standard. The measurements for analytical data were performed on a Finnigan MAT 95 instrument using the fast atom bombardment (FAB) method, where 3-nitrobenzyl alcohol (3-NBA) was used as the matrix. Atmospheric pressure chemical ionization (APCI) and electrospray ionization (ESI) experiments were recorded on a Q-Exactive (Orbitrap) mass spectrometer (Thermo Fisher Scientific, San Jose, CA, USA) equipped with a HESI II probe to record high resolution. Infrared spectra were recorded with an ALPHA-T instrument made by Bruker. Solvents of p.a. quality (*per analysis*) were bought from Sigma Aldrich, Carl Roth or Acros Fisher Scientific and used without previous purification unless otherwise stated.

The experimental details and analytical data for quinones **4b–c** and **5b–c**, anthraquinones **8aa–ac**, acetal **10** as well as [3.3.1]ring system **12** are given below while the experimental data for all other molecules as well as the X-ray analysis can be found in the electronic supplementary material.

## 4.2. General procedures

### 4.2.1. General procedure A for the dienophile precursors (4)

A mixture of trifluoroacetic anhydride (7.00–10.00 equiv.), 1,4-dimethoxynaphthalene (**2**) (1.00 equiv.) and a benzoic acid derivative **3** (1.00–1.20 equiv.) was heated to reflux under argon atmosphere. After 24 h, the mixture was cooled to room temperature, quenched by the addition of H$_2$O and the aqueous phase was extracted with EtOAc. The combined organic phases were washed with saturated aq. NaHCO$_3$ solution, dried over Na$_2$SO$_4$ and the solvents were removed under reduced pressure. The crude product was purified *via* flash chromatography on silica gel.

### 4.2.2. General procedure B for the dienophiles (5)

Under an argon atmosphere, a 1 M solution of ammonium cerium (IV) nitrate (CAN) (3.70 equiv.) in H$_2$O was rapidly added to a 0.1 M solution of the 1,4-dimethoxynaphthalene derivative **4a–g** (1.00 equiv.) in MeCN/CH$_2$Cl$_2$ (4 : 1) at –40°C. The resulting reaction mixture was warmed to –20°C for 1 h and then poured into H$_2$O. The aqueous phase was extracted with EtOAc and the combined organic phases were dried over Na$_2$SO$_4$. The solvents were removed under reduced pressure and the remaining crude product was dissolved in CH$_2$Cl$_2$. CHex was added, and the product was crystallized by the evaporation of CH$_2$Cl$_2$.

### 4.2.3. General procedure C for the Diels–Alder reaction (8/9)

In a crimp vial under an argon atmosphere, the dienophile **5a–g** (1.00 equiv.) was dissolved in dry CH$_2$Cl$_2$ and the diene **6a–h** (3.00–5.00 equiv.) was added. The reaction was stirred at 40°C until the consumption of the dienophile was completed, as indicated by TLC. The solvent was removed under reduced pressure and the crude product was purified *via* flash chromatography on silica gel.

**(1,4-Dimethoxynaphthalen-2-yl)(2-bromophenyl)-methanone (4b):** according to general procedure A, a mixture of trifluoroacetic anhydride (1.5 ml, 1.47 g, 7.00 mmol, 7.00 equiv.), 1,4-dimethoxynaphthalene (**2**) (188 mg, 1.00 mmol, 1.00 equiv.) and 2-bromobenzoic acid (**3b**) (201 mg, 1.00 mmol, 1.00 equiv.) was used. The crude product was purified *via* flash chromatography on silica gel (cHex/EtOAc = 15 : 1). The product **4b** was obtained as a yellow solid (310 mg, 0.835 mmol, 84%).

– $R_f$ (cHex/EtOAc = 15 : 1) = 0.36. – $^1$H NMR (400 MHz, CDCl$_3$): $\delta$ = 8.33–8.25 (m, 1H, C$H_{Ar}$), 8.12–8.05 (m, 1H, C$H_{Ar}$), 7.69–7.64 (m, 1H, C$H_{Ar}$), 7.63–7.53 (m, 2H, C$H_{Ar}$), 7.47 (dd, $^3J$ = 7.5 Hz, $^4J$ = 2.0 Hz, 1H, C$H_{Ar}$), 7.43–7.29 (m, 2H, C$H_{Ar}$), 7.12 (s, 1H, C$H_{Ar}$), 4.04 (s, 3H, OC$H_3$), 3.62 (s, 3H, OC$H_3$) ppm. – $^{13}$C NMR (101 MHz, CDCl$_3$): $\delta$ = 195.3 (C$_q$, 1×C = O), 152.1 (C$_q$, 1×C$_{qAr}$), 151.8 (C$_q$, 1×C$_{qAr}$), 142.3 (C$_q$, 1×C$_{qAr}$), 133.1 (+, 1 × C$H_{Ar}$), 131.1 (+, 1 × C$H_{Ar}$), 129.4 (+, 1 × C$H_{Ar}$), 129.3 (C$_q$, 1 × C$_{qAr}$), 128.5 (C$_q$, 1 × C$_{qAr}$), 128.0 (+, 1 × C$H_{Ar}$), 127.0 (+, 1 × C$H_{Ar}$), 126.8 (+, 1 × C$H_{Ar}$), 126.0 (C$_q$, 1 × C$_{qAr}$), 123.3 (+, 1 × C$H_{Ar}$), 122.4 (+, 1 × C$H_{Ar}$), 119.5 (C$_q$, 1 × C$_{qAr}$), 102.6 (+, 1 × C$H_{Ar}$), 63.8 (+, 1 × OC$H_3$), 55.7 (+, 1 × OC$H_3$) ppm. – IR (ATR): $\nu$ = 2931 (w), 1655 (m), 1583 (m), 1457 (m), 1366 (s), 1271 (m), 1206 (m), 1111 (m), 1092 (m), 1050 (m), 1027 (m), 1000 (m), 955 (m), 860 (m), 846 (m), 804 (m), 761 (m), 750 (s), 692 (m), 669 (m), 652 (m), 636 (m), 480 (w), 458 (w), 429 (w) cm$^{-1}$. – MS (FAB, 3-NBA), m/z (%): 371/373 (82/80) [M+H]$^+$, 370/372 (87/100) [M]$^+$. – HRMS (FAB, C$_{19}$H$_{15}^{79}$BrO$_3$): calc. 370.0205; found 370.0204. – X-ray: the structure of **4b** could be confirmed by single-crystal X-ray diffraction (see crystallographic information in the electronic supplementary material, CCDC 1992178). – repository ID: CRR-9656.

**(2-Bromo-5-methoxyphenyl)(1,4-dimethoxynaphthalen-2-yl)-methanone (4c):** according to general procedure A, a mixture of trifluoroacetic anhydride (2.82 ml, 4.20 g, 20.0 mmol, 10.0 equiv.), 1,4-dimethoxynaphthalene (**2**) (376 mg, 2.00 mmol, 1.00 equiv.) and 2-bromo-5-methoxybenzoic acid (**3c**) (555 mg, 2.40 mmol, 1.20 equiv.) was used. The crude product was purified *via* flash chromatography on silica gel (cHex/EtOAc = 12 : 1). The product **4c** was obtained as a yellow solid (588 mg, 1.47 mmol, 73%). – $R_f$ (cHex/EtOAc = 12 : 1) = 0.24. – $^1$H NMR (400 MHz, CDCl$_3$): $\delta$ = 8.32–8.26 (m, 1H, C$H_{Ar}$), 8.13–8.07 (m, 1H, C$H_{Ar}$), 7.64–7.55 (m, 2H, C$H_{Ar}$), 7.52 (d, $^3J$ = 8.8 Hz, 1H, C$H_{Ar}$), 7.09 (s, 1H, C$H_{Ar}$), 7.00 (d, $^4J$ = 3.0 Hz, 1H, C$H_{Ar}$), 6.90 (dd, $^3J$ = 8.7 Hz, $^4J$ = 3.0 Hz, 1H, C$H_{Ar}$), 4.03 (s, 3H, OC$H_3$), 3.79 (s, 3H, OC$H_3$), 3.67 (s, 3H, OC$H_3$) ppm. – $^{13}$C NMR (101 MHz, CDCl$_3$): $\delta$ = 195.5 (C$_q$, 1 × C = O), 158.8 (C$_q$, 1 × C$_{qAr}$), 152.4 (C$_q$, 1 × C$_{qAr}$), 152.1 (C$_q$, 1 × C$_{qAr}$), 143.4 (C$_q$, 1 × C$_{qAr}$), 134.1 (+, 1 × C$H_{Ar}$), 129.6 (C$_q$, 1 × C$_{qAr}$), 128.8 (C$_q$, 1 × C$_{qAr}$), 128.3 (+, 1 × C$H_{Ar}$), 127.3 (+, 1 × C$H_{Ar}$), 126.2 (C$_q$, 1 × C$_{qAr}$), 123.7 (+, 1 × C$H_{Ar}$), 122.8 (+, 1 × C$H_{Ar}$), 117.5 (+, 1 × C$H_{Ar}$), 114.9 (+, 1 × C$H_{Ar}$), 110.2 (C$_q$, 1 × C$_{qAr}$), 103.0 (+, 1 × C$H_{Ar}$), 64.1 (+, 1 × OC$H_3$), 56.0 (+, 1 × OC$H_3$), 55.8 (+, 1 × OC$H_3$) ppm. – IR (ATR): $\nu$ = 2935 (w), 2838 (w), 1650 (m), 1620 (m), 1592 (m), 1568 (m), 1458 (m), 1404 (m), 1366 (s), 1310 (m), 1281 (m), 1214 (s), 1162 (m), 1113 (m), 1094 (s), 1057 (m), 1020 (m), 962 (m), 853 (m), 813 (m), 767 (s), 709 (m), 669 (m), 603 (m), 486 (w), 430 (w) cm$^{-1}$. – MS (APCI), m/z (%): 401/403 (100/100) [M+H]$^+$. – HRMS (APCI, C$_{20}$H$_{18}^{79}$BrO$_4$): calc. 401.0388; found 401.0374. – X-ray: the structure of **4c** could be confirmed by single-crystal X-ray diffraction (see crystallographic information in the electronic supplementary material, CCDC 1992874). – repository ID: CRR-9668.

**2-(2-Bromobenzoyl)naphthalene-1,4-dione (5b):** following general procedure B, the crude product was obtained from CAN (10.1 g, 18.5 mmol, 3.70 equiv.) and (1,4-dimethoxynaphthalen-2-yl)(2-bromophenyl)methanone (**4b**) (2.09 g, 5.63 mmol, 1.00 equiv.). The product **5b** was isolated as an orange solid (1.92 g, 5.63 mmol, quant.). – $R_f$ (cHex/EtOAc = 9 : 1) = 0.30. – $^1$H NMR (400 MHz, CDCl$_3$): $\delta$ = 8.15–8.08 (m, 2H, C$H_{Ar}$), 7.83–7.79 (m, 2H, C$H_{Ar}$), 7.71–7.69 (m, 1H, C$H_{Ar}$), 7.61 (d, J = 7.8 Hz, 1H, C$H_{Ar}$), 7.48 (t, J = 7.4 Hz, 1H, C$H_{Ar}$), 7.45–7.41 (m, 1H, C$H_{Ar}$), 7.18 (s, 1H, = CH) ppm. – $^{13}$C NMR (101 MHz, CDCl$_3$): $\delta$ = 192.6 (1 × C = O), 185.1 (1 × C = O), 182.7 (1 × C = O), 145.9 (1 × C$_q$), 139.0 (1 × C$_q$), 137.3 (+, 1 × = CH), 134.7 (+, 1 × C$H_{Ar}$), 134.5 (+, 1 × C$H_{Ar}$), 133.9 (+, 1 × C$H_{Ar}$), 133.6 (+, 1 × C$H_{Ar}$), 132.2 (1 × C$_q$), 131.8 (1 × C$_q$), 131.3 (+, 1 × C$H_{Ar}$), 128.0 (+, 1 × C$H_{Ar}$), 127.1 (+, 1 × C$H_{Ar}$), 126.6 (+, 1 × C$H_{Ar}$), 121.0 (1 × C$_q$) ppm. – IR (ATR): $\nu$ = 3041, 1653, 1584, 1465, 1430, 1352, 1328, 1283, 1252, 1101, 1051, 1025, 975, 947, 843, 799, 773, 750, 713, 696, 634, 592, 463, 403 cm$^{-1}$. – MS (FAB, 3-NBA), m/z (%): 342/344 (14/14) [M+H]$^+$, 341/343 (16/21) [M]$^+$. – HRMS (FAB, C$_{17}$H$_{10}^{79}$BrO$_3$): calc. 340.9813; found 340.9814. – X-ray: the structure of **5b** could be confirmed by single-crystal X-ray diffraction (see crystallographic information in the electronic supplementary material, CCDC 1992875). – repository ID: CRR-9662.

**2-(2-Bromo-5-methoxybenzoyl)naphthalene-1,4-dione (5c):** following general procedure B, the crude product was obtained from CAN (10.1 g, 18.4 mmol, 3.70 equiv.) and (2-bromo-5-methoxyphenyl)(1,4-dimethoxynaphthalen-2-yl)methanone (**4c**) (2.00 g, 4.98 mmol, 1.00 equiv.). The product **5c** was isolated as an orange solid (1.37 g, 3.69 mmol, 74%). – $R_f$ (cHex/EtOAc = 4 : 1) = 0.44. – $^1$H NMR (400 MHz, CDCl$_3$): $\delta$ = 8.16–8.08 (m, 2H, C$H_{Ar}$), 7.84–7.78 (m, 2H, C$H_{Ar}$), 7.47 (d, $^3J$ = 8.8 Hz, 1H, C$H_{Ar}$), 7.23 (d, $^4J$ = 3.1 Hz, 1H, C$H_{Ar}$), 7.17 (s, 1H, C = CH), 6.98 (dd, $^3J$ = 8.8 Hz, $^4J$ = 3.1 Hz, 1H, C$H_{Ar}$), 3.86 (s, 3H, OC$H_3$) ppm. – $^{13}$C NMR (101 MHz, CDCl$_3$): $\delta$ = 192.5 (C$_q$, 1 × C = O), 185.1 (C$_q$, 1 × C = O), 182.6 (C$_q$, 1 × C = O), 159.3 (C$_q$, 1 × C$_{qAr}$), 146.1 (C$_q$, 1 × C$_{qAr}$), 139.6 (C$_q$, 1 × C$_{qAr}$), 137.2 (+, 1 × C$H_{Ar}$), 134.7 (+, 1 × C$H_{Ar}$), 134.7 (+, 1 × C$H_{Ar}$), 134.5 (+, 1 × C$H_{Ar}$), 132.2 (C$_q$, 1 × C$_{qAr}$), 131.9 (C$_q$, 1 × C$_{qAr}$), 127.1 (+, 1 × C$H_{Ar}$), 126.6 (+, 1 × C$H_{Ar}$), 120.3 (+, 1 × C$H_{Ar}$), 115.9 (+, 1 × C = CH), 111.6 (C$_q$, 1 × C$_{qAr}$), 55.9 (+, 1 × OC$H_3$) ppm. – IR (ATR): $\tilde{\nu}$ = 2934 (w), 1655 (m), 1589 (m), 1460 (m), 1399

(w), 1349 (w), 1281 (m), 1236 (m), 1097 (w), 1018 (w), 993 (w), 920 (w), 885 (w), 818 (m), 765 (m), 720 (w), 667 (w), 607 (w), 584 (m), 456 (w), 413 (vw) cm$^{-1}$. – MS (FAB, 3-NBA), $m/z$ (%): 371/373 (13/17) [M+H]$^+$, 291 (17), [M–Br]$^+$. – HRMS (FAB, $C_{18}H_{12}^{79}BrO_4$): calc. 370.9919; found 370.9918. – repository ID: CRR-9674.

**4a-(2-Iodobenzoyl)-1,4,4a,9a-tetrahydroanthracene-9,10-dione (8aa)**: a suspension of 3-sulfolene (**7**) (5.00 g, 42.0 mmol, 16.4 equiv.) in $o$-xylene (15 ml) was heated to 125°C for 0.5 h. The thereby developed gaseous 1,3-butadiene (**6a**) was led into a reaction vessel containing a solution of 2-(2-iodobenzoyl)naphthalene-1,4-dione (**5a**) (1.00 g, 3.00 mmol, 1.00 equiv.) in $CH_2Cl_2$ (5.0 ml) at –78°C. After completion of the evolution of 1,3-butadiene gas (**6a**), the mixture of dienophile **5a** and diene **6a** in $CH_2Cl_2$ was slowly warmed to room temperature and stirred at this temperature for 2 h. The solvent was removed under reduced pressure. After flash chromatography on silica gel (cHex/EtOAc = 4 : 1), the product **8aa** was obtained as a colourless solid (925 mg, 2.09 mmol, 81%). – $R_f$ (cHex/EtOAc = 4 : 1) = 0.45. – $^1$H NMR (400 MHz, CDCl$_3$): $\delta$ = 8.17 (dd, $^3J$ = 7.6 Hz, $^4J$ = 1.5 Hz, 1H, CH$_{Ar}$), 8.03 (dd, $^3J$ = 7.8 Hz, $^4J$ = 1.4 Hz, 1H, CH$_{Ar}$), 7.87 (dd, $^3J$ = 7.8 Hz, $^4J$ = 1.1 Hz, 1H, CH$_{Ar}$), 7.76 (dtd, $^3J$ = 23.2, 7.5 Hz, $^4J$ = 1.5 Hz, 2H, CH$_{Ar}$), 7.31 (td, $^3J$ = 7.6 Hz, $^4J$ = 1.1 Hz, 1H, CH$_{Ar}$), 7.16–7.01 (m, 2H, CH$_{Ar}$), 5.70 (s, 2H, CH$_2$-CH = CH-CH$_2$), 3.68 (dd, $^3J$ = 9.9, 6.3 Hz, 1H, C = CH-CH$_2$-CH), 3.07–2.93 (m, 1H, C-CHH-CH), 2.50–2.45 (m, 1H, C-CHH-CH), 2.44–2.38 (m, 1H, CH-CHH-CH), 2.34–2.20 (m, 1H, CH-CHH-CH) ppm. – $^{13}$C NMR (101 MHz, CDCl$_3$): $\delta$ = 200.6 (C$_q$, 1 × $C$ = O), 196.1 (C$_q$, 1 × $C$ = O), 194.0 (C$_q$, 1 × $C$ = O), 142.6 (C$_q$, 1 × C$_{qAr}$), 141.0 (+, 1 × CH$_{Ar}$), 135.3 (+, 1 × CH$_{Ar}$), 134.3 (+, 1 × CH$_{Ar}$), 133.1 (C$_q$, 1 × C$_{qAr}$), 131.7 (+, 1 × CH$_{Ar}$), 127.5 (+, 1 × CH$_{Ar}$), 127.4 (+, 1 × CH$_{Ar}$), 127.2 (+, 1 × CH$_{Ar}$), 126.8 (+, 1 × CH$_{Ar}$), 124.4 (+, 1 × C = CH), 123.9 (+, 1 × C = CH), 93.1 (C$_q$, 1 × C$_{qAr}$), 68.1 (C$_q$, 1 × C$_{qAr}$), 50.0 (+, 1 × CH), 28.4 (–, 1 × CH$_2$), 26.4 (–, 1 × CH$_2$), 14.4 (C$_q$, 1 × C$_q$) ppm. – IR (ATR): $\tilde{v}$ = 2922 (vw), 1690 (w), 1672 (w), 1588 (w), 1424 (vw), 1290 (w), 1248 (w), 1219 (w), 1158 (vw), 1059 (w), 1016 (w), 984 (w), 941 (w), 920 (vw), 891 (vw), 807 (vw), 783 (vw), 761 (w), 746 (w), 728 (w), 687 (w), 660 (w), 635 (vw), 598 (w), 528 (vw), 442 (vw), 407 (w) cm$^{-1}$. – MS (EI, 70 eV), $m/z$ (%): 443 (2) [M+H]$^+$, 442 (7) [M]$^+$, 231 (100) [C$_7$H$_4$IO]$^+$, 211 (26) [C$_{14}$H$_{11}$O$_2$]$^+$, 203 (19) [C$_6$H$_4$I]$^+$. – HRMS (EI, C$_{21}$H$_{15}$IO$_3$): calc. 442.0060; found 442.0062. – X-ray: the structure of **8aa** could be confirmed by single crystal X-ray diffraction (see crystallographic information in the electronic supplementary material, CCDC 1992179). – repository ID: CRR-12148.

**(4aR,9aR)-4a-(2-Bromobenzoyl)-1,4,4a,9a-tetrahydroanthracene-9,10-dione (8ba)**: a suspension of 3-sulfolene (**7**) (2.00 g, 16.9 mmol, 28.9 equiv.) in $o$-xylene (15 ml) was heated to 125°C for 0.5 h. The thereby developed gaseous 1,3-butadiene (**6a**) was then led into a reaction vessel containing a solution of 2-(2-bromobenzoyl)naphthalene-1,4-dione (**5b**) (500 mg, 1.47 mmol, 1.00 equiv.) in $CH_2Cl_2$ (3.0 ml) at –78°C. After completion of the evolution of 1,3-butadiene gas (**6a**), the mixture of dienophile **6b** and diene **6a** in $CH_2Cl_2$ was slowly warmed to room temperature and stirred at this temperature for 2 h. The solvent was removed under reduced pressure. After flash chromatography on silica gel (cHex/EtOAc = 8 : 1), the product **8ba** was obtained as a colourless solid (480 mg, 1.21 mmol, 82%). – $R_f$ (cHex/EtOAc = 8 : 1) = 0.21. – $^1$H NMR (400 MHz, CDCl$_3$): $\delta$ = 8.16 (dd, $^3J$ = 7.6 Hz, $^4J$ = 1.5 Hz, 1H, CH$_{Ar}$), 8.03 (dd, $^3J$ = 7.6 Hz, $^4J$ = 1.4 Hz, 1H, CH$_{Ar}$), 7.79 (td, $^3J$ = 7.6 Hz, $^4J$ = 1.5 Hz, 1H, CH$_{Ar}$), 7.74 (td, $^3J$ = 7.5 Hz, $^4J$ = 1.5 Hz, 1H, CH$_{Ar}$), 7.60–7.55 (m, 1H, CH$_{Ar}$), 7.31–7.22 (m, 2H, CH$_{Ar}$), 7.15–7.09 (m, 1H, CH$_{Ar}$), 5.72–5.64 (m, 2H, CH$_2$-CH = CH-CH$_2$), 3.68 (dd, $^3J$ = 9.6, 6.3 Hz, 1H, C = CH-CH$_2$-CH), 3.03–2.93 (m, 1H, C-CHH-CH), 2.51–2.40 (m, 2H, CHH, CHH), 2.33–2.24 (m, 1H, CHH) ppm. – $^{13}$C NMR (101 MHz, CDCl$_3$): $\delta$ = 199.9 (C$_q$, 1 × $C$ = O), 196.2 (C$_q$, 1 × $C$ = O), 193.9 (C$_q$, 1 × $C$ = O), 139.2 (C$_q$, 1 × C$_{qAr}$), 135.3 (+, 1 × CH$_{Ar}$), 134.3 (+, 1 × CH$_{Ar}$), 134.2 (C$_q$, 1 × C$_{qAr}$), 134.1 (+, 1 × CH$_{Ar}$), 133.1 (C$_q$, 1 × C$_{qAr}$), 131.6 (+, 1 × CH$_{Ar}$), 127.4 (+, 1 × CH$_{Ar}$), 127.2 (+, 2 × CH$_{Ar}$), 126.8 (+, 1 × CH$_{Ar}$), 124.5 (+, 1 × C = CH), 123.8 (+, 1 × C = CH), 119.8 (C$_q$, 1 × C$_{qAr}$), 68.3 (C$_q$, 1 × C$_q$), 49.7 (+, 1 × CH), 28.2 (–, 1 × CH$_2$), 26.2 (–, 1 × CH$_2$) ppm. – IR (ATR): $v$ = 2830 (vw), 2926 (vw), 2885 (vw), 1684 (w), 1589 (w), 1423 (w), 1252 (w), 1218 (w), 1064 (w), 1025 (w), 988 (w), 942 (w), 917 (w), 890 (w), 842 (vw), 798 (vw), 761 (w), 733 (w), 683 (w), 638 (w), 592 (w), 560 (w), 532 (vw), 442 (vw), 424 (vw), 401 (vw) cm$^{-1}$. – MS (FAB, 3-NBA), $m/z$ (%): 395/397 (18/17) [M+H]$^+$. – HRMS (FAB, C$_{21}$H$_{16}^{79}$BrO$_3$): calc. 395.0283; found 395.0285. – X-ray: the structure of **8ba** could be confirmed by single crystal X-ray diffraction (see crystallographic information in the electronic supplementary material, CCDC 1992876). – repository ID: CRR-10340.

**(4aR,9aR)-4a-(2-Iodobenzoyl)-2,3-dimethyl-1,4,4a,9a-tetrahydroanthracene-9,10-dione (8ac):** according to general procedure C, the cycloaddition was performed with 2-(2-iodobenzoyl)naphthalene-1,4-dione (**5a**) (388 mg, 1.00 mmol, 1.00 equiv.) and 2,3-dimethylbuta-1,3-diene (**6c**) (0.34 ml, 246 mg, 3.00 mmol, 3.00 equiv.) in dry $CH_2Cl_2$ (5.0 ml). After 3 h, the crude product was purified $via$ flash chromatography on silica gel (cHex/EtOAc = 9 : 1) to obtain product **8ac** as a yellow solid (328 mg, 697 μmol, 70%). – $R_f$ (cHex/EtOAc = 9 : 1) = 0.31. – $^1$H NMR (400 MHz, CDCl$_3$): $\delta$ = 8.16 (dd, $^3J$ = 7.6 Hz, $^4J$ = 1.5 Hz, 1H, CH$_{Ar}$), 8.02

(dd, $^3J$ = 7.5 Hz, $^4J$ = 1.6 Hz, 1H, C$H_{Ar}$), 7.88 (dd, $^3J$ = 8.1 Hz, $^4J$ = 1.2 Hz, 1H, C$H_{Ar}$), 7.78 (td, $^3J$ = 7.5 Hz, $^4J$ = 1.5 Hz, 1H, C$H_{Ar}$), 7.73 (td, $^3J$ = 7.5 Hz, $^4J$ = 1.5 Hz, 1H, C$H_{Ar}$), 7.33 (td, $^3J$ = 7.6 Hz, $^4J$ = 1.1 Hz, 1H, C$H_{Ar}$), 7.22 (dd, $^3J$ = 7.8 Hz, $^4J$ = 1.7 Hz, 1H, C$H_{Ar}$), 7.08 (td, $^3J$ = 7.7 Hz, $^4J$ = 1.7 Hz, 1H, C$H_{Ar}$), 3.70 (t, $^3J$ = 7.7 Hz, 1H, C$H$), 2.81 (d, $^2J$ = 17.2 Hz, 1H, C$HH$), 2.38 (d, $^2J$ = 17.2 Hz, 1H, C$HH$), 2.29 (d, $^3J$ = 7.7 Hz, 2H, C$H_2$), 1.60 (s, 3H, C$H_3$), 1.55 (s, 3H, C$H_3$) ppm. – $^{13}$C NMR (101 MHz, CDCl$_3$): $\delta$ = 200.9 (C$_q$, 1 × C = O), 196.4 (C$_q$, 1 × C = O), 194.8 (C$_q$, 1 × C = O), 142.6 (C$_q$, 1 × C$_{qAr}$), 141.0 (+, 1 × C$H_{Ar}$), 135.2 (+, 1 × C$H_{Ar}$), 134.4 (C$_q$, 1 × C$_{qAr}$), 134.3 (+, 1 × C$H_{Ar}$), 133.2 (C$_q$, 1 × C$_{qAr}$), 131.6 (+, 1 × C$H_{Ar}$), 127.4 (+, 1 × C$H_{Ar}$), 127.4 (+, 1 × C$H_{Ar}$), 127.2 (+, 1 × C$H_{Ar}$), 127.1 (+, 1 × C$H_{Ar}$), 123.9 (C$_q$, 1 × C$_q$), 123.1 (C$_q$, 1 × C$_q$), 93.2 (C$_q$, 1 × C$_{qAr}$), 68.5 (C$_q$, 1 × C$_q$), 50.5 (+, 1 × CH), 34.6 (–, 1 × CH$_2$), 31.9 (–, 1 × CH$_2$), 19.1 (+, 1 × CH$_3$), 18.9 (+, 1 × CH$_3$) ppm. – IR (ATR): $\tilde{v}$ = 3065 (vw), 2912 (vw), 1678 (w), 1592 (w), 1425 (w), 1252 (w), 1218 (w), 1055 (vw), 1005 (w), 933 (vw), 884 (vw), 836 (vw), 764 (w), 745 (w), 672 (vw), 638 (vw), 609 (vw), 554 (vw), 447 (vw), 386 (vw) cm$^{-1}$. – MS (FAB, 3-NBA), $m/z$ (%): 470 (3) [M]$^+$, 471 (16) [M+H]$^+$, 307 (32), 231 (40), 154 (100). – HRMS (FAB, C$_{23}$H$_{20}^{127}$IO$_3$): calc. 471.0457; found 471.0456. – X-ray: the structure of **8ac** could be confirmed by single-crystal X-ray diffraction (see crystallographic information in the electronic supplementary material, CCDC 1992877). – repository ID: CRR-9796.

**(4aR,9aR)-4a-(2-Iodobenzoyl)-2,2-dimethoxy-1,2,3,4,4a,9a-hexahydroanthracene-9,10-dione (10):** to a solution of (4aR,9aR)-2-((*tert*-butyldiphenylsilyl)oxy)-4a-(2-iodobenzoyl)-1,4,4a,9a-tetrahydroanthracene-9,10-dione (**8ah**) (44.6 mg, 64.0 μmol, 1.00 equiv.) in a mixture of dry MeOH and dry CH$_2$Cl$_2$ (1 : 1) (0.4 ml) was added a drop of AcCl (50 μl, 55.3 mg, 700 μmol, 11 equiv.) at 0°C and the reaction mixture was stirred for 3.5 h at this temperature. After completion of the reaction (monitored by TLC), CH$_2$Cl$_2$ was added (4.0 ml), the reaction mixture was neutralized with 10% NaHCO$_3$ (0.5 ml) and washed with H$_2$O (5 ml). The organic layer was dried over Na$_2$SO$_4$ and concentrated *in vacuo* to give the crude product, which was purified *via* silica gel flash chromatography on silica gel (cHex/EtOAc = 3 : 1) to give product **10** (31.0 mg, 61.5 μmol, 96%). – $R_f$ (cHex/EtOAc = 3 : 1) = 0.37. – $^1$H NMR (300 MHz, CDCl$_3$): $\delta$ = 8.21–8.15 (m, 1H, C$H_{Ar}$), 8.08–8.02 (m, 1H, C$H_{Ar}$), 7.85–7.68 (m, 3H, C$H_{Ar}$), 7.29 (td, $^3J$ = 7.6 Hz, $^4J$ = 1.2 Hz, 1H, C$H_{Ar}$), 7.07 (td, $^3J$ = 7.7 Hz, $^2J$ = 1.6 Hz, 1H, C$H_{Ar}$), 6.91 (dd, $^3J$ = 7.7 Hz, $^4J$ = 1.6 Hz, 1H, C$H_{Ar}$), 3.63 (dd, $J$ = 13.9, 4.1 Hz, 1H, C$H$), 3.21 (s, 3H, OC$H_3$), 3.14 (s, 3H, OC$H_3$), 2.69–2.58 (m, 1H, C$HH$), 2.27 (ddd, $J$ = 13.6, 4.1, 3.0 Hz, 1H, C$HH$), 2.10–1.97 (m, 1H, C$HH$), 1.85 (td, $J$ = 13.4, 4.1 Hz, 1H, C$HH$), 1.67–1.53 (m, 1H, C$HH$), 1.40 (t, $^2J$ = 13.8 Hz, 1H, C$HH$) ppm. – $^{13}$C NMR (76 MHz, CDCl$_3$): $\delta$ = 201.0 (C$_q$, 1 × C = O), 195.8 (C$_q$, 1 × C = O), 193.1 (C$_q$, 1 × C = O), 143.6 (C$_q$, 1 × C$_{qAr}$), 140.2 (+, 1 × C$H_{Ar}$), 135.3 (C$_q$, 1 × C$_{qAr}$), 134.3 (+, 1 × C$H_{Ar}$), 134.0 (C$_q$, 1 × C$_{qAr}$), 133.4 (+, 1 × C$H_{Ar}$), 131.2 (+, 1 × C$H_{Ar}$), 127.3 (+, 3 × C$H_{Ar}$), 126.1 (+, 1 × C$H_{Ar}$), 98.2 (C$_q$, 1 × C$_q$OCH$_3$), 91.5 (C$_q$, 1 × C$_{qAr}$), 69.4 (C$_q$, 1 × C$_q$), 51.6 (+, 1 × CH), 47.9 (+, 1 × OCH$_3$), 47.7 (+, 1 × OCH$_3$), 34.9 (–, 1 × CH$_2$), 28.7 (–, 1 × CH$_2$), 26.3 (–, 1 × CH$_2$) ppm. – IR (ATR): $\tilde{v}$ = 395 (w), 424 (w), 445 (m), 487 (w), 561 (w), 596 (w), 636 (m), 674 (w), 691 (s), 732 (vs), 744 (vs), 759 (s), 772 (s), 785 (m), 807 (w), 823 (m), 846 (m), 902 (m), 929 (vs), 982 (s), 1006 (s), 1045 (vs), 1081 (vs), 1112 (s), 1143 (vs), 1160 (w), 1218 (vs), 1255 (vs), 1358 (w), 1425 (w), 1459 (w), 1562 (w), 1591 (m), 1680 (vs), 2830 (w), 2936 (w), 2949 (w) cm$^{-1}$. – MS (EI, 70 eV), $m/z$ (%): 504 (12) [M]$^+$, 473 (28) [M–OCH$_3$]$^+$, 231 (100). – HRMS (EI, C$_{23}$H$_{22}$O$_3^{127}$I): calc. 473.0614; found 473.0616. – repository ID: CRR-12066.

**(5S,13aR)-6H-5,13a-Methanobenzo[4,5]cycloocta[1,2-b]naphthalene-8,13,14(5H)-trione** (**12**): under argon atmosphere, a mixture of (4aR,9aR)-4a-(2-iodobenzoyl)-1,4,4a,9a-tetrahydroanthracene-9,10-dione (**8aa**) (53.1 mg, 120 μmol, 1.00 equiv.), Pd(OAc)$_2$ (5.4 mg, 24.0 μmol, 20 mol%), PPh$_3$ (12.6 mg, 48.0 μmol, 40 mol%) was placed into a high-pressure glass tube. The mixture was dissolved in dry dimethylacetamide (0.50 ml), pentamethylpiperidine (45 μl, 37.2 mg, 240 μmol, 2.00 equiv.) was added and the mixture was stirred at 70°C for 22 h. After completion of the reaction, as indicated by TLC, it was quenched by the addition of H$_2$O. The aqueous phase was extracted with EtOAc and the combined organic phases were dried over Na$_2$SO$_4$. The solvents were removed under reduced pressure and the crude product was purified *via* flash chromatography on silica gel (cHex/EtOAc = 6 : 1) to obtain **12** as a yellow solid (27.0 mg, 85.9 μmol, 72%). – $R_f$ (cHex/EtOAc = 6 : 1) = 0.22. – $^1$H NMR (400 MHz, CDCl$_3$): $\delta$ = 8.25–8.22 (m, 1H, C$H_{Ar}$), 8.18–8.15 (m, 1H, C$H_{Ar}$), 7.88–7.84 (m, 1H, C$H_{Ar}$), 7.82–7.78 (m, 1H, C$H_{Ar}$), 7.56 (td, $^3J$ = 7.5 Hz, $^4J$ = 1.5 Hz, 1H, C$H_{Ar}$), 7.34–7.29 (m, 2H, C$H_{Ar}$), 7.24 (m, 2H, C=C$H$, C$H_{Ar}$), 3.58 (m, 1H, C$H$), 3.16 (ddd, $^2J$ = 13.5 Hz, $^3J$ = 3.8 Hz, $^4J$ = 1.7 Hz, 1H, C$HH$), 3.01 (ddd, $^2J$ = 20.6 Hz, $^3J$ = 6.1, 2.7 Hz, 1H, C$HH$), 2.52 (ddt, $^2J$ = 20.6 Hz, $^3J$ = 5.2 Hz, $^4J$ = 1.6 Hz, 1H, C$HH$), 2.28 (dd, $^2J$ = 13.5 Hz, $^3J$ = 2.6 Hz, 1H, C$HH$) ppm. – $^{13}$C NMR (101 MHz, CDCl$_3$): $\delta$ = 196.2 (C$_q$, 1 × C = O), 195.1 (C$_q$, 1 × C = O), 183.0 (C$_q$, 1 × C = O), 147.1 (C$_q$, 1 × C$_{qAr}$), 140.2 (+, 1 × C$H_{Ar}$), 137.3 (C$_q$, 1 × C$_{qAr}$), 135.4 (C$_q$, 1 × C$_{qAr}$), 135.1 (+, 1 × C$H_{Ar}$), 134.9 (+, 1 × C$H_{Ar}$), 134.8 (+, 1 × C$H_{Ar}$), 132.7 (C$_q$, 1 × C$_{qAr}$), 129.4 (C$_q$, 1 × C$_{qAr}$), 128.8 (+, 1 × C$H_{Ar}$), 128.7 (+, 1 × C$H_{Ar}$), 128.1 (+, 2 × C$H_{Ar}$), 127.0 (+, 1 × C = CH), 59.8 (C$_q$, 1 × C$_q$), 34.3 (–, 1 × CH$_2$), 32.5 (+, 1 × CH), 31.1 (–, 1 × CH$_2$) ppm. – IR (ATR): $\tilde{v}$ = 2923 (w), 1731 (m), 1697 (m), 1669 (m), 1589 (m), 1453 (w), 1411 (w), 1269 (s), 1250 (s),

1156 (m), 969 (w), 946 (w), 927 (m), 890 (w), 858 (m), 825 (w), 780 (w), 754 (m), 718 (m), 694 (m), 635 (w), 578 (w), 541 (w), 519 (w), 445 (w) cm$^{-1}$. – MS (EI, 70 eV), $m/z$ (%): 314 (91) [M+H]$^{+}$. – HRMS (EI, C$_{21}$H$_{14}$O$_3$): calc. 314.0937; found 314.0937. – X-ray: the structure of **12** could be confirmed by single-crystal X-ray diffraction (see crystallographic information in the electronic supplementary material, CCDC 1992893). – repository ID: CRR-12176.

Data accessibility. The obtained data were deposited in the repository Chemotion (reaction details and compound characterization) and the CCDC (crystal structures). The related IDs which can be used to identify the submissions (web access: https://www.chemotion-repository.net/home/publications; https://www.ccdc.cam.ac.uk/structures/) are given as repository ID (Chemotion Repository Reaction ID—CRR; Cambridge Crystallographic Data Centre—CCDC). Chemotion Repository: https://dx.doi.org/10.14272/reaction/SA-FUHFF-UHFFFADPSC-VIFKGORYWD-UHFFFADPSC-NUHFF-NUHFF-NUHFF-ZZZ **(4a)**; https://dx.doi.org/10.14272/reaction/SA-FUHFF-UHFFFADPSC-BLDZZLLBER-UHFFFADPSC-NUHFF-NUHFF-NUHFF-ZZZ **(4b)**; https://dx.doi.org/10.14272/reaction/SA-FUHFF-UHFFFADPSC-ISSAVWGYAU-UHFFFADPSC-NUHFF-NUHFF-NUHFF-ZZZ **(4c)**; https://dx.doi.org/10.14272/reaction/SA-FUHFF-UHFFFADPSC-ANVMBMPZHQ-UHFFFADPSC-NUHFF-NUHFF-NUHFF-ZZZ **(4d)**; https://dx.doi.org/10.14272/reaction/SA-FUHFF-UHFFFADPSC-OPAWPZCMGG-UHFFFADPSC-NUHFF-NUHFF-NUHFF-ZZZ **(4e)**; https://dx.doi.org/10.14272/reaction/SA-FUHFF-UHFFFADPSC-IOGOYECUJC-UHFFFADPSC-NUHFF-NUHFF-NUHFF-ZZZ **(4f)**; https://dx.doi.org/10.14272/reaction/SA-FUHFF-UHFFFADPSC-CIZDEMMLYC-UHFFFADPSC-NUHFF-NUHFF-NUHFF-ZZZ **(5a)**; https://dx.doi.org/10.14272/reaction/SA-FUHFF-UHFFFADPSC-UANJEQXEUP-UHFFFADPSC-NUHFF-NUHFF-NUHFF-ZZZ **(5b)**; https://dx.doi.org/10.14272/reaction/SA-FUHFF-UHFFFADPSC-HDMBZXQPNM-UHFFFADPSC-NUHFF-NUHFF-NUHFF-ZZZ **(5c)**; https://dx.doi.org/10.14272/reaction/SA-FUHFF-UHFFFADPSC-LQLVWTVLQD-UHFFFADPSC-NUHFF-NUHFF-NUHFF-ZZZ **(5d)**; https://dx.doi.org/10.14272/reaction/SA-FUHFF-UHFFFADPSC-BCJAVVNVCV-UHFFFADPSC-NUHFF-NUHFF-NUHFF-ZZZ **(5e)**; https://dx.doi.org/10.14272/reaction/SA-FUHFF-UHFFFADPSC-MWUPABMFGF-UHFFFADPSC-NUHFF-NUHFF-NUHFF-ZZZ **(5f)**; https://dx.doi.org/10.14272/reaction/SA-FUHFF-UHFFFADPSC-YREQDOIZJY-UHFFFADPSC-NUHFF-NUHFF-NUHFF-ZZZ **(5g)**; https://dx.doi.org/10.14272/reaction/SA-FUHFF-UHFFFADPSC-TZBXDZKIGZ-UHFFFADPSC-NUHFF-NDRCZ-NUHFF-ZZZ **(8aa)**; https://dx.doi.org/10.14272/reaction/SA-FUHFF-UHFFFADPSC-BGJIBRAFUH-UHFFFADPSC-NUHFF-NDRCZ-NUHFF-ZZZ **(8ba)**; https://dx.doi.org/10.14272/reaction/SA-FUHFF-UHFFFADPSC-HTUXSVMUHU-UHFFFADPSC-NUHFF-NUWWY-NUHFF-ZZZ **(8/9ab)**; https://dx.doi.org/10.14272/reaction/SA-FUHFF-UHFFFADPSC-UWSTXCXPGI-UHFFFADPSC-NUHFF-NUWWY-NUHFF-ZZZ **(8/9bb)**; https://dx.doi.org/10.14272/reaction/SA-FUHFF-UHFFFADPSC-BJHRMKGCWP-UHFFFADPSC-NUHFF-NUGKD-NUHFF-ZZZ **(8ac)**; https://dx.doi.org/10.14272/reaction/SA-FUHFF-UHFFFADPSC-VQHSHKAJCS-UHFFFADPSC-NUHFF-NUHFF-NUHFF-ZZZ **(8bc)**; https://dx.doi.org/10.14272/reaction/SA-FUHFF-UHFFFADPSC-ZUBFTCRJYO-UHFFFADPSC-NUHFF-NRFRK-NUHFF-ZZZ **(8ad, 9ad)**; https://dx.doi.org/10.14272/reaction/SA-FUHFF-UHFFFADPSC-LRHPRUPFRY-UHFFFADPSC-NUHFF-NRFRK-NUHFF-ZZZ **(8bd, 9bd)**; https://dx.doi.org/10.14272/reaction/SA-FUHFF-UHFFFADPSC-PHEQJDFJHL-UHFFFADPSC-NUHFF-NUWAF-NUHFF-ZZZ **(8ae, 9ae)**; https://dx.doi.org/10.14272/reaction/SA-FUHFF-UHFFFADPSC-UMSRHQGWSN-UHFFFADPSC-NUHFF-NHVYL-NUHFF-ZZZ **(8af, 9af)**; https://dx.doi.org/10.14272/reaction/SA-FUHFF-UHFFFADPSC-AYOHXYGZVK-UHFFFADPSC-NUHFF-NHVYL-NUHFF-ZZZ **(8/9bf)**; https://dx.doi.org/10.14272/reaction/SA-FUHFF-UHFFFADPSC-GYGDJDFWRC-UHFFFADPSC-NUHFF-NXQUO-NUHFF-ZZZ **(8ag)**; https://dx.doi.org/10.14272/reaction/SA-FUHFF-UHFFFADPSC-VGCVCLYBTD-UHFFFADPSC-NUHFF-NLYQR-NUHFF-ZZZ **(8ah)**; https://dx.doi.org/10.14272/reaction/SA-FUHFF-UHFFFADPSC-FOIXCRBFVL-UHFFFADPSC-NUHFF-NLYQR-NUHFF-ZZZ **(8bh)**; https://dx.doi.org/10.14272/reaction/SA-FUHFF-UHFFFADPSC-XQBNRFKTLB-UHFFFADPSC-NUHFF-NUGKD-NUHFF-ZZZ **(8/9bb)**; https://dx.doi.org/10.14272/reaction/SA-FUHFF-UHFFFADPSC-UKTGPOJWCW-UHFFFADPSC-NUHFF-NUHFF-NUHFF-ZZZ **(8cc)**; https://dx.doi.org/10.14272/reaction/SA-FUHFF-UHFFFADPSC-TWNQMZQUUQ-UHFFFADPSC-NUHFF-NNMJJ-NUHFF-ZZZ **(8/9cd)**; https://dx.doi.org/10.14272/reaction/SA-FUHFF-UHFFFADPSC-PWQZDRACEZ-UHFFFADPSC-NUHFF-NGJAF-NUHFF-ZZZ **(8/9db)**; https://dx.doi.org/10.14272/reaction/SA-FUHFF-UHFFFADPSC-FDOSUKYERH-UHFFFADPSC-NUHFF-NUHFF-NUHFF-ZZZ **(8dc)**; https://dx.doi.org/10.14272/reaction/SA-FUHFF-UHFFFADPSC-GNLDMLVTJD-UHFFFADPSC-NUHFF-NLDCB-NUHFF-ZZZ **(8ec)**; https://dx.doi.org/10.14272/reaction/SA-FUHFF-UHFFFADPSC-YMLGBYYTIH-UHFFFADPSC-NUHFF-NRQYZ-NUHFF-ZZZ **(8/9fb)**; https://dx.doi.org/10.14272/reaction/SA-FUHFF-UHFFFADPSC-BBXGNCLCSR-UHFFFADPSC-NUHFF-NUHFF-NUHFF-ZZZ **(8fc)**; https://dx.doi.org/10.14272/reaction/SA-FUHFF-UHFFFADPSC-CYVQCLLXOP-UHFFFADPSC-NUHFF-NGMWY-NUHFF-ZZZ **(8/9gb)**; https://dx.doi.org/10.14272/reaction/SA-FUHFF-UHFFFADPSC-LWPRQCGBQO-UHFFFADPSC-NUHFF-NUHFF-NUHFF-ZZZ **(8gc)**; https://dx.doi.org/10.14272/reaction/SA-FUHFF-UHFFFADPSC-SMNBKPYPJX-UHFFFADPSC-NUHFF-NDPHN-NUHFF-ZZZ **(8/9gd)**; https://dx.doi.org/10.14272/reaction/SA-FUHFF-UHFFFADPSC-UXHJHTBPRQ-UHFFFADPSC-NUHFF-NUHFF-NUHFF-ZZZ **(10)**; https://dx.doi.org/10.14272/reaction/SA-FUHFF-UHFFFADPSC-IKYAJDXLFU-UHFFFADPSC-NUHFF-NFYFD-NUHFF-ZZZ **(12)**. Cambridge Crystallographic Data Centre (CCDC): 1992178 **(4b)**, 1992179 **(8aa)**, 1992180 **(5a)**, 1992181 **(9ad)**, 1992182 **(8ae)**, 1992183 **(9af)**, 1992184 **(8af)**, 1992185 **(8bh)**, 1992874 **(4c)**, 1992875 **(5b)**, 1992876 **(8ba)**, 1992877 **(8ac)**; 1992878 **(4a)**, 1992879 **(4d)**, 1992880 **(4e)**, 1992881 **(4f)**, 1992882 **(5d)**, 1992883 **(8bc)**, 1992884 **(8ad)**, 1992885 **(9bd)**, 1992886 **(8bd)**, 1992887 **(9ae)**, 1992888 **(9bf)**, 1992889 **(8ag)**, 1992890 **(8fc)**, 1992891 **(8gc)**, 1992892 **(9gd)** and 1992893 **(12)** contain the supplementary crystallographic data for this paper.

These data can be obtained free of charge from The Cambridge Crystallographic Data Centre via www.ccdc.cam.ac.uk/data_request/cif.

Authors' contributions. J.B. carried out the laboratory work, analysed most of the obtained data and wrote the manuscript under the supervision and guidance of S.B. O.F. and M.N. performed X-ray experiments and analysed the obtained data. All authors gave final approval for publication.

Competing interests. There are no conflicts to declare.

Funding. J.B. gratefully acknowledges financial support from the Carl-Zeiss Stiftung.

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
