## [Reviewer comments · Royal Society Open Science]

Review History

RSOS-200626.R0 (Original submission)

Review form: Reviewer 1

Is the manuscript scientifically sound in its present form?

Yes

Are the interpretations and conclusions justified by the results?

Yes

Is the language acceptable?

Yes

Do you have any ethical concerns with this paper?

No

Have you any concerns about statistical analyses in this paper?

No

Recommendation?

Accept with minor revision (please list in comments)

Comments to the Author(s)

In the manuscript "A versatile Diels–Alder Approach to Functionalized Hydroanthraquinones", the authors report a large body of preparative chemistry of the cycloaddition reactions between 2-benzoylbenzoquinones and electron-rich 1,3-dienes to give the corresponding benzoyl-substituted hydroanthraquinones in good to excellent yields with good selectivities. This is inspired by the structural features of some hydroanthraquinone-containing natural products that exhibit interesting bioactivities. Most notably each demonstrated product contains a quaternary stereogenic carbon center, and the authors also give an example of further functionalization of the Diels–Alder product to furnish an elaborated [3.3.1]-ring system. The current study will be suitable for publication at Royal Society Open Science provided that the authors kindly address my comments below:

1. For the scheme at the top of Table 1, I think the authors mis-assigned the endo vs. exo products. Compounds 8 series should be the exo products and compounds 9 series should be the endo products. The endo/exo terminology refers to the relative spatial orientation of the diene-dienophile relationship in the Diels–Alder transition state structure, not the relative configuration of the stereogenic centers in the products. Therefore, the entire discussion involving endo/exo in the following section needs to be flipped (for example, a big portion on page 5).
2. Correcting the endo/exo terminology would explain why the hindered dienes 6d, 6e, and 6f still participate the Diels–Alder reaction smoothly, because in the endo transition state, the bulky siloxy group is actually pointed away from the bulky benzoyl group on the dienophile. I do agree with the authors in that the Diels–Alder reactions are governed by the stereoelectronic nature of the reactants (which derives the endo/exo phenomenon) instead of steric effects. When there's overwhelming steric hindrance, the reaction is simply shut down. I think it would serve as an indirect argument to support that the formation of the endo products do not suffer from steric hindrance because there is less steric clash in the endo transition state than in the exo transition state (because of the presence of the bulky benzoyl group).
3. The authors use the term "ortho" and "para" a few times to denote the regioselectivity of the Diels–Alder reaction. Although educated readers will understand based on the basics of [4+2]-cycloaddition and the schemes in the manuscript, I think it will be less ambiguous if the authors could number the hydroanthraquinone skeleton for the relevant structures and refer to the position of the substituents by the carbon numbers.
4. On page 6, I think it's unnecessary and abrupt for the authors to reiterate the rationale for the synthesis of 5c-5g while they are addressing the performances of the reactions in Table 1. The authors have already shown these substrates earlier on page 2. I would prefer that they provide the rationale when they first introduced these substrates.
5. Just curious when diene 6d or 6e are used, how stable are the corresponding products against silica gel?
6. The authors comment that reagent such as TBAF would decompose their hydroanthraquinone compounds. I can see with compounds such as 8bd and 9bd, with treatment of TBAF, the resulting alkoxide would undergo retro-Alder reaction as a possible decomposition pathway. I'm curious if the authors have tried any other milder silyl group removal conditions. Even anhydrous HCl (which is the actual catalyst under the AcCl/MeOH condition) is a little harsh.
7. Just wondering if the bromo-compound 8ba would also undergo the Heck reaction in comparable yield to give 12?
8. I think the authors' discussion regarding the results in Table 1 is a little too lengthy. Many of those reactions behave similarly. They could group the dienes into several categories and consolidate the commentary more efficiently.

Review form: Reviewer 2

Is the manuscript scientifically sound in its present form?

Yes

Are the interpretations and conclusions justified by the results?

Yes

Is the language acceptable?

Yes

Do you have any ethical concerns with this paper?

No

Have you any concerns about statistical analyses in this paper?

No

Recommendation?

Accept with minor revision (please list in comments)

Comments to the Author(s)

The authors report a strategy and methodology for the synthesis of a class of natural products, anthraquinones, that are of very high importance given their inspiration and use for antibiotics and chemotherapies. The authors tackle a particularly vexing problem in this work: that of quaternary carbon centers that are often present.

The Diels-Alder is an incredibly powerful and oft-used reaction, but, as E.J. Corey once said, we have only scratched the surface of the utility and power of the Diels-Alder reaction.

I note the use of several functional groups that demonstrate the tolerances of the reaction sequence: amines, ethers, and hydroxyls.

The yields and regioselectivity are useful in a synthetic context. The diastereoselectivity is usually practical, though a few examples show low diastereoselectivity. The authors do show a very wide matrix of diene/dienophile combinations to demonstrate the power of their strategy. This array of structures, many confirmed crystallographically, provides evidence of the product structures with confidence.

Lastly, the authors disclose two post-diels Alder transformations: the functional group interconversion of a silyl enol ether to a dimethyl acetal and, more impressively, a Heck reaction to form a bicyclo[3.3.1]nonane ring that is close to the structure of the ortho-beticolins. While finding selective Heck reaction conditions for a bicyclo[3.2.2]nonane product are highly unlikely given the endo-like geometry required, I do think they will be able to find an alternative strategy.

Overall, this report details an accessible approach to an important structural core for antibiotics, chemotherapies, and natural products that those who desire these compounds may utilize.

Two minor recommendations are noted:

Table 1, footnote a says: "Absolute stereochemistry was determined by X-ray diffraction." I understand these reactions to be non-enantioselective, so a single enantiomer with one absolute stereochemistry is not appropriate. I believe this statement should be "Relative stereochemistry was determined by X-ray diffraction" if I am understanding correctly.

Also in Table 1, it would help the reader to designate the R3, R4, and R5 groups in the products 8 and 9 to show the regioisomer produced from non-symmetric dienes, especially isoprene 6b. The authors describe the product in the text (including the exact isomer obtained), and they show exact isomers in the SI, but clarification in the table will also be helpful.

Review form: Reviewer 3 (Michael Krout)

Is the manuscript scientifically sound in its present form?

Yes

Are the interpretations and conclusions justified by the results?

Yes

Is the language acceptable?

Yes

Do you have any ethical concerns with this paper?

No

Have you any concerns about statistical analyses in this paper?

No

Recommendation?

Accept as is

Comments to the Author(s)

Overall this is a very nice, thorough study detailing the use of a DA reaction to prepared a number of complex hydroanthroquinones. The work details a number of dienophiles and dienes in a survey of the regioselective and diastereoselective (endo/exo) outcome of the DA reaction. The detailed experimental and characterization data will of immense use to anyone that wishes to replicate the work. The authors explore some new chemical space with this reaction and appropriately reference Brimble's work toward this carbon framework (ref 18). It would therefore contribute to the chemical literature.

On page three of manuscript (has Scheme 3 at the top), I would recommend changing the second sentence of the first paragraph (line 17). It states that "two new stereocenters are created" when there were none to begin with. I'd remove "new" since there are two stereocenters that form.

All DA reactions were performed thermally. Have the authors tried to use Lewis or Brønsted acid catalysis? It would be helpful to see this mentioned (not necessarily include data) as this is the most common way to accelerate a DA reaction. This might also give a sense of asymmetric DA possibilities.

Finally, page 6 second column of the manuscript describes an effort to cleave the TMS enol ether. TBAF provided decomposition and HCl (AcCl/MeOH) provided the dimethyl acetal. I have experienced some weird side reactions with TBAF myself and they were likely due to hydroxide (or fluoride) cause some base-induced reactions. My solution was to buffer the TBAF with a AcOH in a 1:1 ratio in THF. The rate of cleavage decreases, but reactions can be warmed to mitigate the slowing of rate. Just a thought/suggestion.

Overall this is nice work and should be published as is.

Decision letter (RSOS-200626.R0)

Dear Dr Braese:

Title: A Versatile Diels-Alder Approach to Functionalized Hydroanthraquinones
Manuscript ID: RSOS-200626

Thank you for submitting the above manuscript to Royal Society Open Science. On behalf of the Editors and the Royal Society of Chemistry, I am pleased to inform you that your manuscript will be accepted for publication in Royal Society Open Science subject to minor revision in accordance with the referee suggestions. Please find the reviewers' comments at the end of this email. I apologise it has taken longer than usual to send you this decision.

The reviewers and handling editors have recommended publication, but also suggest some minor revisions to your manuscript. Therefore, I invite you to respond to the comments and revise your manuscript.

Because the schedule for publication is very tight, it is a condition of publication that you submit the revised version of your manuscript before 20-Sep-2020. Please note that the revision deadline will expire at 00.00am on this date. If you do not think you will be able to meet this date please let me know immediately.

Kind regards,
Dr Laura Smith
Publishing Editor, Journals

On behalf of the Subject Editor Professor Anthony Stace and the Associate Editor Dr Andrew Harned.

RSC Associate Editor:

Comments to the Author:

The referees are united in presenting strong enthusiasm for this manuscript and I join them in this assessment. This paper was a pleasure to read and will become a welcome addition to the synthetic literature after the authors address the minor concerns raised by the reviewers.

RSC Subject Editor:

Comments to the Author:

(There are no comments.)

Reviewer comments to Author:

Reviewer: 1

Comments to the Author(s)

In the manuscript "A versatile Diels-Alder Approach to Functionalized Hydroanthraquinones", the authors report a large body of preparative chemistry of the cycloaddition reactions between 2-benzoylbenzoquinones and electron-rich 1,3-dienes to give the corresponding benzoyl-substituted hydroanthraquinones in good to excellent yields with good selectivities. This is inspired by the structural features of some hydroanthraquinone-containing natural products that exhibit interesting bioactivities. Most notably each demonstrated product contains a quaternary stereogenic carbon center, and the authors also give an example of further functionalization of the Diels-Alder product to furnish an elaborated [3.3.1]-ring system. The current study will be suitable for publication at Royal Society Open Science provided that the authors kindly address my comments below:

1. For the scheme at the top of Table 1, I think the authors mis-assigned the endo vs. exo products. Compounds 8 series should be the exo products and compounds 9 series should be the endo products. The endo/exo terminology refers to the relative spacial orientation of the diene-dienophile relationship in the Diels-Alder transition state structure, not the relative configuration of the stereogenic centers in the products. Therefore, the entire discussion involving endo/exo in the following section needs to be flipped (for example, a big portion on page 5).
2. Correcting the endo/exo terminology would explain why the hindered dienes 6d, 6e, and 6f still participate the Diels-Alder reaction smoothly, because in the endo transition state, the bulky siloxy group is actually pointed away from the bulky benzoyl group on the dienophile. I do agree

with the authors in that the Diels–Alder reactions are governed by the stereoelectronic nature of the reactants (which derives the endo/exo phenomenon) instead of steric effects. When there's overwhelming steric hindrance, the reaction is simply shut down. I think it would serve as an indirect argument to support that the formation of the endo products do not suffer from steric hindrance because there is less steric clash in the endo transition state than in the exo transition state (because of the presence of the bulky benzoyl group).

3. The authors use the term "ortho" and "para" a few times to denote the regioselectivity of the Diels–Alder reaction. Although educated readers will understand based on the basics of [4+2]-cycloaddition and the schemes in the manuscript, I think it will be less ambiguous if the authors could number the hydroanthraquinone skeleton for the relevant structures and refer to the position of the substituents by the carbon numbers.

4. On page 6, I think it's unnecessary and abrupt for the authors to reiterate the rationale for the synthesis of 5c-5g while they are addressing the performances of the reactions in Table 1. The authors have already shown these substrates earlier on page 2. I would prefer that they provide the rationale when they first introduced these substrates.

5. Just curious when diene 6d or 6e are used, how stable are the corresponding products against silica gel?

6. The authors comment that reagent such as TBAF would decompose their hydroanthraquinone compounds. I can see with compounds such as 8bd and 9bd, with treatment of TBAF, the resulting alkoxide would undergo retro-Alder reaction as a possible decomposition pathway. I'm curious if the authors have tried any other milder silyl group removal conditions. Even anhydrous HCl (which is the actual catalyst under the AcCl/MeOH condition) is a little harsh.

7. Just wondering if the bromo-compound 8ba would also undergo the Heck reaction in comparable yield to give 12?

8. I think the authors' discussion regarding the results in Table 1 is a little too lengthy. Many of those reactions behave similarly. They could group the dienes into several categories and consolidate the commentary more efficiently.

Reviewer: 2

Comments to the Author(s)

The authors report a strategy and methodology for the synthesis of a class of natural products, anthraquinones, that are of very high importance given their inspiration and use for antibiotics and chemotherapies. The authors tackle a particularly vexing problem in this work: that of quaternary carbon centers that are often present.

The Diels-Alder is an incredibly powerful and oft-used reaction, but, as E.J. Corey once said, we have only scratched the surface of the utility and power of the Diels-Alder reaction.

I note the use of several functional groups that demonstrate the tolerances of the reaction sequence: amines, ethers, and hydroxyls.

The yields and regioselectivity are useful in a synthetic context. The diastereoselectivity is usually practical, though a few examples show low diastereoselectivity. The authors do show a very wide matrix of diene/dienophile combinations to demonstrate the power of their strategy. This array of structures, many confirmed crystallographically, provides evidence of the product structures with confidence.

Lastly, the authors disclose two post-diels Alder transformations: the functional group interconversion of a silyl enol ether to a dimethyl acetal and, more impressively, a Heck reaction to form a bicyclo[3.3.1]nonane ring that is close to the structure of the ortho-beticolins. While finding selective Heck reaction conditions for a bicyclo[3.2.2]nonane product are highly unlikely given the endo-like geometry required, I do think they will be able to find an alternative strategy.

Overall, this report details an accessible approach to an important structural core for antibiotics, chemotherapies, and natural products that those who desire these compounds may utilize.

Two minor recommendations are noted:

Table 1, footnote a says: "Absolute stereochemistry was determined by X-ray diffraction." I understand these reactions to be non-enantioselective, so a single enantiomer with one absolute stereochemistry is not appropriate. I believe this statement should be "Relative stereochemistry was determined by X-ray diffraction" if I am understanding correctly.

Also in Table 1, it would help the reader to designate the R3, R4, and R5 groups in the products 8 and 9 to show the regioisomer produced from non-symmetric dienes, especially isoprene 6b. The authors describe the product in the text (including the exact isomer obtained), and they show exact isomers in the SI, but clarification in the table will also be helpful.

Reviewer: 3

Comments to the Author(s)

Overall this is a very nice, thorough study detailing the use of a DA reaction to prepare a number of complex hydroanthroquinones. The work details a number of dienophiles and dienes in a survey of the regioselective and diastereoselective (endo/exo) outcome of the DA reaction. The detailed experimental and characterization data will of immense use to anyone that wishes to replicate the work. The authors explore some new chemical space with this reaction and appropriately reference Brimble's work toward this carbon framework (ref 18). It would therefore contribute to the chemical literature.

On page three of manuscript (has Scheme 3 at the top), I would recommend changing the second sentence of the first paragraph (line 17). It states that "two new stereocenters are created" when there were none to begin with. I'd remove "new" since there are two stereocenters that form.

All DA reactions were performed thermally. Have the authors tried to use Lewis or Brønsted acid catalysis? It would be helpful to see this mentioned (not necessarily include data) as this is the most common way to accelerate a DA reaction. This might also give a sense of asymmetric DA possibilities.

Finally, page 6 second column of the manuscript describes an effort to cleave the TMS enol ether. TBAF provided decomposition and HCl (AcCl/MeOH) provided the dimethyl acetal. I have experienced some weird side reactions with TBAF myself and they were likely due to hydroxide (or fluoride) cause some base-induced reactions. My solution was to buffer the TBAF with a AcOH in a 1:1 ratio in THF. The rate of cleavage decreases, but reactions can be warmed to mitigate the slowing of rate. Just a thought/suggestion.

Overall this is nice work and should be published as is.

Author's Response to Decision Letter for (RSOS-200626.R0)

See Appendix A.

Decision letter (RSOS-200626.R1)

Dear Dr Braese:

Title: A Versatile Diels-Alder Approach to Functionalized Hydroanthraquinones
Manuscript ID: RSOS-200626.R1

It is a pleasure to accept your manuscript in its current form for publication in Royal Society Open Science. The chemistry content of Royal Society Open Science is published in collaboration with the Royal Society of Chemistry.

On behalf of the Subject Editor Professor Anthony Stace and the Associate Editor Dr Andrew Harned.

RSC Associate Editor
Comments to the Author:

The referees have addressed all of the concerns raised by the previous reviewers, and the manuscript can now be published. I look forward to seeing this work in print.

Reviewer(s)' Comments to Author:

Appendix A

Response to Referees

Title: A Versatile Diels-Alder Approach to Functionalized Hydroanthraquinones

Manuscript ID: RSOS-200626

Dear Dr. Laura Smith

We would like to thank the Editorial Office and reviewers for their time to evaluate our manuscript for publication in Royal Society Open Science. We highly appreciate all the comments from the Editorial Office and the reviewers' valuable suggestions and thank for the appreciation of this work. We believe that including these suggestions greatly contributed to improving our manuscript, for general readership and experts working in this particular area of hydroanthraquinone synthesis.

In compliance with the reviewers' comments and suggestions, we have revised the manuscript. Please find below our point-by-point answers and responses to the reviewer's comments and suggestions. The reviewers' comments are reproduced unchanged with our response included in italics.

Thank you very much for your kind consideration.
Looking forward to hearing from you soon.

With best regards,

Stefan Bräse

Institute of Organic Chemistry (IOC), Karlsruhe Institute of Technology (KIT)

Point-by-point response to the Editorial Office and Reviewers

Reviewer comments to Author:

Reviewer: 1

In the manuscript "A versatile Diels-Alder Approach to Functionalized Hydroanthraquinones", the authors report a large body of preparative chemistry of the cycloaddition reactions between 2-benzoylbenzoquinones and electron-rich 1,3-dienes to give the corresponding benzoyl-substituted hydroanthraquinones in good to excellent yields with good selectivities. This is inspired by the structural features of some hydroanthraquinone-containing natural products that exhibit interesting bioactivities. Most notably each demonstrated product contains a quaternary stereogenic carbon center, and the authors also give an example of further functionalization of the Diels-Alder product to furnish an elaborated [3.3.1]-ring system. The current study will be suitable for publication at Royal Society Open Science provided that the authors kindly address my comments below:

We thank the reviewer for the support of our manuscript and the positive comments. In compliance with the reviewer's comments and suggestions, we have revised the manuscript.

1. For the scheme at the top of Table 1, I think the authors mis-assigned the endo vs. exo products.

Compounds 8 series should be the exo products and compounds 9 series should be the endo products. The endo/exo terminology refers to the relative spacial orientation of the diene-dienophile relationship in the Diels–Alder transition state structure, not the relative configuration of the stereogenic centers in the products. Therefore, the entire discussion involving endo/exo in the following section needs to be flipped (for example, a big portion on page 5).

We agree with the reviewer comments and therefore now corrected the assignment of exo and endo products.

2. Correcting the endo/exo terminology would explain why the hindered dienes 6d, 6e, and 6f still participate the Diels–Alder reaction smoothly, because in the endo transition state, the bulky siloxy group is actually pointed away from the bulky benzoyl group on the dienophile. I do agree with the authors in that the Diels–Alder reactions are governed by the stereoelectronic nature of the reactants (which derives the endo/exo phenomenon) instead of steric effects. When there's overwhelming steric hindrance, the reaction is simply shut down. I think it would serve as an indirect argument to support that the formation of the endo products do not suffer from steric hindrance because there is less steric clash in the endo transition state than in the exo transition state (because of the presence of the bulky benzoyl group).

Following the reviewer's suggestion, we have now included the proposed argument in the manuscript to support that the formation of the endo products do not suffer from steric hindrance.

3. The authors use the term "ortho" and "para" a few times to denote the regioselectivity of the Diels–Alder reaction. Although educated readers will understand based on the basics of [4+2]-cycloaddition and the schemes in the manuscript, I think it will be less ambiguous if the authors could number the hydroanthraquinone skeleton for the relevant structures and refer to the position of the substituents by the carbon numbers.

To enhance clarity concerning the ortho/para nomenclature, we included the sentence "The regioselectivity of the Diels-Alder reactions is described by the ortho/meta/para nomenclature with 1,2-disubstituted adducts named "ortho" as well as 1,4-adducts referred to as "para"." and numbered the hydroanthraquinone skeleton accordingly.

4. On page 6, I think it's unnecessary and abrupt for the authors to reiterate the rational for the synthesis of 5c-5g while they are addressing the performances of the reactions in Table 1. The authors have already shown these substrates earlier on page 2. I would prefer that they provide the rational when they first introduced these substrates.

We thank the reviewer for the suggestion. The rational for the synthesis of 5c-5g is now removed from page 6.

5. Just curious when diene 6d or 6e are used, how stable are the corresponding products against silica gel?

The products were purified via standard column chromatography, however one needed to be quick to prevent deprotection of the trimethylsilyl ether.

6. The authors comment that reagent such as TBAF would decompose their hydroanthraquinone compounds. I can see with compounds such as 8bd and 9bd, with treatment of TBAF, the resulting alkoxide would undergo retro-Alder reaction as a possible decomposition pathway. I'm curious if the authors have tried any other milder silyl group removal conditions. Even anhydrous HCl (which is the actual catalyst under the AcCl/MeOH condition) is a little harsh.

We also tried to use 1M HCl in CH₂Cl₂ diluted in MeOH as well as 1% TFA in MeOH (+CH₂Cl₂) which both did not yield the desired products.

7. Just wondering if the bromo-compound 8ba would also undergo the Heck reaction in comparable yield to give 12?

Bromo-compound 8ba also undergoes the Heck reaction towards 12, however it results in a decreased yield of 59% in comparison to the iodinated precursor 8aa.

8. I think the authors' discussion regarding the results in Table 1 is a little too lengthy. Many of those reactions behave similarly. They could group the dienes into several categories and consolidate the commentary more efficiently.

Regarding the limited amount of time given for the revision, unfortunately we were not able to restructure the discussion. We believe that a thorough discussion of the cycloaddition reactions is a substantial part of the manuscript, and therefore kindly ask the reviewer to accept the discussion as is.

Reviewer: 2

The authors report a strategy and methodology for the synthesis of a class of natural products, anthraquinones, that are of very high importance given their inspiration and use for antibiotics and chemotherapies. The authors tackle a particularly vexing problem in this work: that of quaternary carbon centers that are often present.

The Diels-Alder is an incredibly powerful and oft-used reaction, but, as E.J. Corey once said, we have only scratched the surface of the utility and power of the Diels-Alder reaction.

I note the use of several functional groups that demonstrate the tolerances of the reaction sequence: amines, ethers, and hydroxyls.

The yields and regioselectivity are useful in a synthetic context. The diastereoselectivity is usually practical, though a few examples show low diastereoselectivity. The authors do show a very wide matrix of diene/dienophile combinations to demonstrate the power of their strategy. This array of structures, many confirmed crystallographically, provides evidence of the product structures with confidence.

Lastly, the authors disclose two post-diels Alder transformations: the functional group interconversion of a silyl enol ether to a dimethyl acetal and, more impressively, a Heck reaction to form a bicyclo[3.3.1]nonane ring that is close to the structure of the ortho-beticolins. While finding selective Heck reaction conditions for a bicyclo[3.2.2]nonane product are highly unlikely given the endo-like geometry required, I do think they will be able to find an alternative strategy.

Overall, this report details an accessible approach to an important structural core for antibiotics, chemotherapies, and natural products that those who desire these compounds may utilize.

We greatly appreciate the reviewer's very kind remarks and appreciation of this work.

Two minor recommendations are noted:

Table 1, footnote a says: "Absolute stereochemistry was determined by X-ray diffraction." I understand these reactions to be non-enantioselective, so a single enantiomer with one absolute stereochemistry is not appropriate. I believe this statement should be "Relative stereochemistry was determined by X-ray diffraction" if I am understanding correctly.

We agree with the reviewer and therefore now corrected the statement in the footnote of Table 1.

Also in Table 1, it would help the reader to designate the R3, R4, and R5 groups in the products 8 and 9 to show the regioisomer produced from non-symmetric dienes, especially isoprene 6b. The authors describe the product in the text (including the exact isomer obtained), and they show exact isomers in the SI, but clarification in the table will also be helpful.

According to the ortho/para rule, only ortho and para products were obtained, as shown in the scheme of Table 1. Also, in Table 1 the dienes are drawn in the exact same orientation as they react with the dienophile and therefore the residues at the hydroanthraquinone skeleton are oriented as shown in the dienes. To enhance clarity, we described the ortho/para nomenclature in the manuscript and numbered the hydroanthraquinone skeleton.

Reviewer: 3

Overall this is a very nice, thorough study detailing the use of a DA reaction to prepared a number of complex hydroanthroquinones. The work details a number of dienophiles and dienes in a survey of the regioselective and diastereoselective (endo/exo) outcome of the DA reaction. The detailed experimental and characterization data will of immense use to anyone that wishes to replicate the work. The authors explore some new chemical space with this reaction and appropriately reference Brimble's work toward this carbon framework (ref 18). It would therefore contribute to the chemical literature.

We thank the reviewer for the support of our manuscript and the positive comments.

On page three of manuscript (has Scheme 3 at the top), I would recommend changing the second sentence of the first paragraph (line 17). It states that "two new stereocenters are created" when there were none to begin with. I'd remove "new" since there are two stereocenters that form.

We agree with the reviewer and therefore now removed the "new".

All DA reactions were performed thermally. Have the authors tried to use Lewis or Brønsted acid catalysis? It would be helpful to see this mentioned (not necessarily include data) as this is the most common way to accelerate a DA reaction. This might also give a sense of asymmetric DA possibilities.

*Yes, we have tried that. The application of $TiCl_4$ as well as the application of $ZnCl_2$ resulted in the decomposition of iodinated naphthoquinone **5a**. When $B(OAc)_3$ was used in the reaction of iodo dienophile **5a** with TBDMS diene **6f**, the yield of the reaction was increased to 78% while the ratio of isomers did not change. The addition of $BF_3 \cdot OEt_2$ to the cycloaddition between dienophile **5a** and TMS diene **6d** at $-20^\circ C$ resulted in a decreased total yield of 52% without affecting the isomeric ratio of products.*

*Through the addition of Schreiner's catalyst in the cycloaddition of naphthoquinone **5a** with TBDMS diene **6f** the reaction time was shortened to 2 h with the yield as well as the ratio of isomers not being affected. The application of Jacobsen's catalyst resulted in a slightly increased selectivity towards endo product **9af** (ratio **8af** to **9af** 1 : 1.4) with consistent yield. It thus was assumed that both organocatalysts do not form a complex with dienophile **5a** in a way that enables significant stereoselectivity of the reaction.*

Since all of these attempts were not satisfying and thermal acceleration enabled smooth cycloadditions, we decided to not include the above described results in the manuscript.

Finally, page 6 second column of the manuscript describes an effort to cleave the TMS enol ether. TBAF provided decomposition and HCl (AcCl/MeOH) provided the dimethyl acetal. I have experienced some weird side reactions with TBAF myself and they were likely due to hydroxide (or fluoride) cause some base-induced reactions. My solution was to buffer the TBAF with a AcOH in a 1:1 ratio in THF. The rate of cleavage decreases, but reactions can be warmed to mitigate the slowing of rate. Just a thought/suggestion.

We greatly appreciate the valuable suggestion and will give it a try in future experiments.

Overall this is nice work and should be published as is.

We would like to take the opportunity to acknowledge the reviewers' valuable suggestions. Their sincere efforts in reviewing the manuscript are much appreciated. We believe that including the reviewers' comments and suggestions have greatly contributed to improving our manuscript.

Thank you once again for your kind consideration.

With best regards,

Stefan Bräse

Institute of Organic Chemistry (IOC), Karlsruhe Institute of Technology (KIT)